

# Tropospheric ozone trends and attributions over East and Southeast Asia in 1995-2019: An integrated assessment using statistical methods, machine learning models, and multiple chemical transport models

Xiao Lu[1,2*#], Yiming Liu[1,2#], Jiayin Su[1,2#], Xiang Weng[3#], Tabish Ansari[4#], Yuqiang Zhang[5#], Guowen He[1,2], Yuqi Zhu[1,2], Haolin Wang[1,2], Ganquan Zeng[1,2], Jingyu Li[1,2], Cheng He[1,2], Shuai Li[1,2], Teerachai Amnuaylojaroen[6], Tim Butler[4], Qi Fan[1,2], Shaojia Fan[1,2], Grant L. Forster[3,7], Meng Gao[8], Jianlin Hu[9], Yugo Kanaya[10], Mohd Talib Latif[11], Keding Lu[12], Philippe Nédélec[13], Peer Nowack[14,15], Bastien Sauvage[13], Xiaobin Xu[16], Lin Zhang[17], Ke Li[9], Ja-Ho Koo[18], Tatsuya Nagashima[19]

[1]School of Atmospheric Sciences, Sun Yat-sen University,  Zhuhai, 519082, China.
[2]Guangdong Provincial Observation and Research Station for Climate Environment and Air Quality Change in the Pearl River Estuary, Southern Marine Science and Engineering Guangdong Laboratory (Zhuhai), Zhuhai, 519082, China.
[3]School of Environmental Sciences, University of East Anglia, Norwich, NR47 TJ, UK
[4]Research Institute for Sustainability – Helmholtz Centre Potsdam (RIFS), Potsdam,  Germany
[5]Environment Research Institute, Shandong University, Qingdao 266237, China
[6]Department of Environmental Science, School of Energy and Environment, University of Phayao, Phayao, Thailand
[7]National Centre for Atmospheric Science, University of East Anglia, Norwich, UK
[8]Department of Geography, Hong Kong Baptist University, Hong Kong SAR, China
[9]Jiangsu Key Laboratory of Atmospheric Environment Monitoring and Pollution Control, Collaborative Innovation Center of Atmospheric Environment and Equipment Technology, School of Environmental Science and Engineering, Nanjing University of Information Science and Technology, Nanjing, 210044, China
[10]Earth Surface System Research Center, Research Institute for Global Change, Japan Agency for Marine-Earth Science and Technology (JAMSTEC), Yokohama, Kanagawa, 2360001, Japan
[11]Department of Earth Sciences and Environment, Faculty of Science and Technology, Universiti Kebangsaan Malaysia, Bangi, Malaysia
[12]State Key Joint Laboratory of Environmental Simulation and Pollution Control, College of Environmental Sciences and Engineering, Peking University, Beijing, 100871, China
[13]Laboratoire d'Aérologie, CNRS and Université Toulouse III, Paul Sabatier, Toulouse, France
[14]Institute of Theoretical Informatics, Karlsruhe Institute of Technology, 76131 Karlsruhe, Germany
[15]Institute of Meteorology and Climate Research (IMK-ASF), Karlsruhe Institute of Technology, 76131 Karlsruhe, Germany
[16]Institute of Atmospheric Composition, Chinese Academy of Meteorological Sciences, Beijing 100081, China
[17]Laboratory for Climate and Ocean-Atmosphere Sciences, Department of Atmospheric and Oceanic Sciences, School of Physics, Peking University, Beijing 100871, China
[18]Department of Atmospheric Sciences, Yonsei University, Seoul 03722, South Korea
[19]National Institute for Environmental Studies, Tsukuba, 305-8506, Japan
#These authors contributed equally to this work.

*Correspondence to*: Xiao Lu (luxiao25@mail.sysu.edu.cn)

**Abstract.** We apply a statistical model, two machine learning models, and three chemical transport models to attribute the observed ozone increases over East and Southeast Asia (ESEA) to changes in anthropogenic emissions and climate. Despite variations in model capabilities and emission inventories, all chemical transport models agree that increases in anthropogenic emission are a primary driver of ozone increases in 1995-2019.The models attribute 53-59% of the increase in tropospheric



ozone burden over ESEA to changes in anthropogenic emissions, with emission within ESEA contributing by 66-77%. South Asia has increasing contribution to ozone increases over ESEA. At the surface, the models attribute 69-75% of the ozone increase in 1995-2019 to changes in anthropogenic emissions. Climate change also contributes substantially to the increase in summertime tropospheric (41-47%) and surface ozone (25-31%). We find that emission reductions in China since 2013 have

led to contrasting responses in ozone levels in the troposphere (decrease) and at the surface (increase). From 2013 to 2019, the ensemble mean derived from multiple models estimate that 66% and 56% of the summertime surface ozone enhancement in the North China Plain and the Yangtze River Delta could be attributed to changes in anthropogenic emissions, respectively, with the remaining attributed to meteorological factors. In contrast, changes in anthropogenic emissions dominate summertime ozone increase in the Pearl River Delta and Sichuan Basin (about 95%). Our study underscores the need for long-term

observational data, improved emission inventories, and advanced modeling frameworks to better understand the mechanisms of ozone increases in ESEA.

## 1 Introduction

Ozone plays a crucial role in the atmosphere as a major oxidant and a short-lived greenhouse gas. At ground level, ozone poses

significant risks to human health, harms vegetation, and reduces crop yields (Monks et al., 2015). Ozone in the troposphere is chemically produced from nitrogen oxides ($NO_x$), carbon monoxide (CO), and volatile organic compounds (VOCs) in the presence of sunlight. Transport from the stratosphere is another source of tropospheric ozone. Since the preindustrial era, tropospheric ozone burden has risen by 45%, contributing to a global effective radiative forcing of 0.47 (0.24 to 0.70) W m$^{-2}$ (including stratospheric and tropospheric ozone, 1750-2019), with a continuous increase since the 1990s (IPCC, AR6).

Ozone is growing especially fast over the densely populated regions of East and Southeast Asia (ESEA). Analysis from aircraft observations from the In-service Aircraft for a Global Observing System database (IAGOS) demonstrates increase in tropospheric ozone column (950-250hPa) at a rate of 2.5-5.0 ppbv decade$^{-1}$ from 1995 to 2017 in these areas (Gaudel et al., 2020; Wang et al., 2022a). This increase rate is among the highest when compared to other regions in the Northern Hemisphere,

with even more substantial growth observed in the lower troposphere (below 850hPa). The reported increasing trends derived from IAGOS observations are consistent with ozonesonde observations in Japan (Wang et al., 2022a), and Beijing (Zhang et al., 2020), Hong Kong (Wang et al.,2019) in China. They also align with trends derived from satellite products (Ziemke et al., 2019; Gaudel et al., 2020). The ozone increase in the lower troposphere over Southeast Asia can significantly impact global tropospheric chemistry and ozone distribution, through frequent deep convection and subsequent atmospheric circulations

(Lawrence and Lelieveld, 2010; Lu et al., 2018b).



At ground level, present-day ozone concentrations in East Asia (including China, Japan, and Korean Peninsula) have been shown to be distinctly higher than those in the US and Europe, as reported by the Tropospheric Ozone Assessment Report Phase I (TOAR I) and subsequent studies (Gaudel et al., 2018; Lu et al., 2018a; 2020; Lyu et al., 2023). Both Japan and South
Korea have documented substantial surface ozone increases since the 1990s (Akimoto et al., 2015; Seo et al., 2014; Nagashima et al., 2017; Yeo and Kim, 2020; Kim et al., 2023). For instance, Kim et al. (2023) reported an ozone increase across all urban and background sites from 2000 to 2021. However, a recent study demonstrates that the increase rate in warm-season daily maximum 8-hour average (MDA8) ozone in Japan and South Korea decelerated after 2010 compared to the preceding decades (Wang et al., 2024). Long-term surface ozone measurement is relatively scarce in China. Several studies have reported notable
ozone increases at background sites in the North China Plain and the Pearl River Delta, moderate increases at the global baseline site in western China, and decreases at a site in northwestern China (Sun et al., 2016; Ma et al., 2016; Xu et al., 2016; Wang et al., 2019; Xu et al., 2020). The national network established in 2013 to monitor air quality in major Chinese cities has recorded a significant rise in April-September MDA8 ozone, with an increase of 2.4 ppbv year$^{-1}$ from 2013 to 2019 (Lu et al., 2020). This surge occurs despite substantial reductions in anthropogenic $NO_x$ emissions. Surface observations have also
documented ozone increases over Peninsular Southeast Asia and the Maritime Continents (Wang et al., 2022b). For example, studies have shown notable enhancement in ozone concentrations ranging from 0.09 to 0.21 ppbv year$^{-1}$ during 1997-2016 at four sites in the western Peninsular Malaysia (Latif et al., 2016; Ahamad et al., 2020).

Quantification of the underlying causes of ozone increases is essential for developing effective ozone mitigation strategies in
ESEA. Tropospheric ozone trends are driven by variations in anthropogenic emissions of its precursors, and are also influenced by climate change, which modulates ozone by affecting the natural sources, photochemistry, and transport of ozone even in the absence of trends in anthropogenic emissions (Lu et al., 2019a; Fiore et al., 2022). On a global scale, studies have revealed the dominant role of shifts in anthropogenic emissions, including contributions from aircraft emissions and background methane, in tropospheric ozone increases since 1980 (Zhang et al., 2016; Wang et al., 2022a). Factors contributing to ozone
changes in US and Europe are extensively studied and quantified (e.g., Lin et al., 2017; Yan et al., 2018). However, in ESEA where rapid tropospheric ozone growth is occurring, it remains largely unquantitative to what extent ozone trends at different times and spatial scales can be attributed to trends in anthropogenic emissions of ozone precursors within or outside of ESEA, and to climate change. This limitation partly arises from the scarcity of observations for constraining the long-term trends, uncertainties in emission inventory, and the computational costs for conducting chemical model simulations over multiple
decades. Specifically, in attributing the post-2013 surface ozone trends in China, modeling studies have revealed significant discrepancies in how these trends are linked to changes in anthropogenic emissions and meteorological conditions, both in their direction and magnitude (Li et al., 2020; Liu and Wang et al., 2020a,b; Dang et al., 2021; Weng et al., 2022; Liu et al., 2023). These discrepancies reflect the differences in modeling approaches (statistical models, machine learning methods, versus three-dimensional chemical transport models), model capabilities (chemical mechanisms and resolution), input data
(meteorological and emission), and time frames, which can confuse the attribution of ozone trends.



Building upon the observational basis and identified scientific gaps, the East Asia Working Group in TOAR Phase II is dedicated to exploring three key questions: (1) What are the spatiotemporal distributions and trends of tropospheric ozone in East Asia, (2) What drives tropospheric ozone trends over East Asia, and (3) How does ozone change over ESEA influences
downwind ozone air quality, global ozone budgets, and other atmospheric constituents?

This study aims to answer the second question. Acknowledging the surging ozone level in Southeast Asia, our study expands the spatial coverage to include both East and Southeast Asia (Fig.1). We apply diverse methodologies to attribute long-term and short-term ozone trends, spanning from the surface to the tropopause across ESEA. We focus on the boreal summertime
(June, July, and August) when most regions in East Asia shows peak ozone concentrations in the year, but it does not cover the ozone season in some regions in Southeast Asia where ozone typically peaks in boreal autumn or winter (e.g., Latif et al., 2016). We choose 1995-2019 for the long-term trends analysis, as ozone measurements in the free troposphere over ESEA become increasingly accessible in this period. For short-term trends, we focus on the years from 2013 to 2019, as nationwide surface ozone measurement in China starts from 2013 and shows significant ozone increase in this period. A distinctive feature
of this study is the integrated application of a statistical model, two machine learning models, and three chemical transport models, each with unique characteristics, to evaluate the consistency and discrepancies among these models in reproducing current levels and trends in tropospheric ozone across East Asia. This enables us to quantify the uncertainty in ozone trend attributions, taking into account the variations in model capabilities, meteorological inputs, and emissions data. In relation to this study, a companion paper (Li et al., summited) delves deeper into the contemporary levels and trends of ozone from the
surface to the tropopause over ESEA (Question 1), and a separate study will examine the implications of tropospheric ozone and its precursors over ESEA on the global atmospheric chemistry (Question 3).

The study is structured as follows: Section 2 outlines the observations and models applied in this work. Section 3 briefly examines trends in meteorological parameters, anthropogenic and natural sources of ozone precursors. Section 4 evaluates the
capability of chemical models to reproduce current ozone levels, spatial patterns, and trends. Section 5 attributes the factors influencing long-term (1995-2019) and short-term (2013-2019) ozone trends. Conclusions and discussions are summarized in Section 6.



## 2 Observations, model descriptions, and experiment design

### 2.1 Observational data

**2.1.1 Surface measurement network**

We collect hourly ozone observations from 1995 to 2019 from national surface monitoring networks and individual sites, covering major developed and developing regions in ESEA (Figure 1a, Table 1). The national monitoring network is from China, Japan, South Korea, Malaysia, and Thailand. We analyze the data in Japan (1995-2019), South Korea (2001-2019), Hong Kong China (2001-2019), Thailand (2005-2019), mainland China (2013-2019), and Malaysia (2017). Five additional

monitoring sites in China with more than 11 years of available observations are also included (Xu et al., 2020). Detailed descriptions of the monitoring networks and data quality control measures will be given in the companion paper (Li et al., in prep.).

### 2.1.2 Ozonesonde observations

We utilize measurements of vertical ozone profiles from seven ozonesonde sites documented in the World Ozone and

Ultraviolet Radiation Data Centre (WOUDC) (Fig.1a). The four sites in East Asia, i.e., Pohang, Naha, Sapporo, and Tateno (Tsukuba), record over 230 ozone profiles during June, July, and August from 1995 to 2019, with an average of approximately 3-4 profiles per month. In contrast, the three sites in Southeast Asia have significantly fewer available ozone profiles of around 100 per site, except for King's Park site in Hong Kong, China. Here, we categorize the seven stations into two groups (East Asia and Southeast Asia), ensuring an adequate number of data samples to better characterize the tropospheric ozone profiles

and trends in both regions.

### 2.1.3 IAGOS observations

We apply measurements of tropospheric ozone profiles in 1995-2019 from the In-service Aircraft for a Global Observing System database (IAGOS) program. The IAGOS program was initiated in 1994 (Thouret et al., 1998) to measure multiple atmospheric compositions including ozone, with instruments on board commercial aircraft (Nédélec et al., 2015). Details for

the measurements and validation are extensively documented in previous studies (Thouret et al., 1998; Nédélec et al., 2015; Blot et al., 2021). Measurements of tropospheric ozone are available during takeoff and landing, and during the cruise portion of the flight at any time of the day. The sampling frequency varies depending on the airline schedule. Figure 1b summarizes available IAGOS profiles over East Asia (103°-137°E, 27°-46°N, N=2584) and Southeast Asia (90°-123°E, -10°-27°N, N=2059). The IAGOS flight height typically reaches up to 200 hPa, which can attain or exceed the tropopause in East Asia,

yet remains within the upper troposphere in tropical Southeast Asia. Analyses of IAGOS data indicate their consistency with ozonesonde records in the upper troposphere–lower stratosphere above Western Europe (Staufer et al., 2013) and their representation of ozone in the lower troposphere (Petetin et al., 2018; Cooper et al., 2020). The IAGOS data has been applied



to derive robust tropospheric ozone trends on a regional scale from the northern mid-latitudes to the tropics (Cohen et al., 2018; Cooper et al., 2020; Gaudel et al., 2020; Wang et al., 2022a).

## 2.2 Statistical and machine learning models

Figure S1 provides an overview of the application of three statistical and machine learning models used in this study for attribution of ozone trends. The overall strategy is to apply these approaches to develop a predictive model for surface ozone concentrations using meteorological variables, and from which to separately quantify the role of meteorology and other factors (ideally linked to emissions) in ozone variability and trends. We adopt one conventional statistical method, i.e., the multiple linear regression (MLR) method, and two machine learning models, i.e., the ridge regression (RR) and random forest regression (RFR) methods.

We use the stepwise MLR modeling approach. For each step, the model selects the most powerful metrological predictor explaining the residual of ozone variability, re-examines the model with the new predictor included, and removes existing predictors with insignificant influence if necessary. This process continues until only the three most statistically significant predictors left. MLR then models only relying on these three predictors, thereby reducing potential collinearity and the risk of overfitting that are often associated with conventional MLR, in which all predictors are considered. The ridge regression in this study is a linear regression in essence, but with its cost function augmented with L2-regularization, which can also effectively improve collinearity and overfitting in conventional MLR (McDonald, 2009). Unlike stepwise MLR, RR—because of its L2-regularization—can achieve resistance to collinearity and overfitting, but without the need for eliminating predictors. RFR, on the other hand, is an ensemble decision tree approach that can adaptively model both linear and nonlinear relationships between predictors and the dependent variable (Breiman, 2001; Grange et al., 2018). These three models have been applied to assess the contributions of meteorology and emission on ozone variabilities in China (Li et al., 2019b, 2020; Weng et al., 2022). Our application here, with consistent time frame, and data process (e.g., deseasonalization, as documented below), allows a direct comparison of results from the three approaches.

We obtain the meteorological variables from the fifth-generation European Centre for Medium-Range Weather Forecasts atmospheric reanalysis of the global climate (ERA5, horizontal resolution of 0.25°×0.25°, latitude × longitude) and Modern Era Retrospective analysis for Research and Application version 2 (MERRA2, 0.5°×0.625°) reanalysis datasets in turn. Specifically, 11 meteorological variables (Table S1) are selected from each dataset (i.e., ERA5 and MERRA2 in turn) as predictors for these algorithms to model ozone. The selected meteorological variables include temperature, solar radiation, wind speed, and others that have been widely recognized to modulate daily ozone variability (Li et al., 2019b; Gong and Liao, 2019; Weng et al., 2022; Yang et al., 2024). We perform the analyses at the city level by averaging ozone concentrations across monitoring sites within the same city to represent the average air quality of that city. To spatially align the gridded meteorological data with the in-situ MDA8 measurements, we extract the meteorological data from the reanalysis datasets at



the grid point corresponding to the city center. Following Weng et al. (2022), we then deseasonalize both MDA8 ozone and daytime (06:00-18:00 local time) meteorological data by subtracting the multi-year averaged 15-days moving mean window from each corresponding data point, based on the same date in month-day format. This step is to prevent ozone predictions from being influenced by inherent seasonality rather than daily variability in meteorology. Finally, these 11 deseasonalized

meteorological variables serve as predictors fed into MLR, RR, and RFR to predict the dependent variable, namely, deasonalized surface MDA8 ozone.

We follow standard machine learning practices by splitting the training and testing sets for RR and RFR. Specifically, the entire dataset is randomly split into two parts, a training set comprising 80% of the data and the rest 20% for testing. We utilize

a two-stage five-fold random partition method. The first stage, as mentioned above, is designed to randomly partition the dataset, with each of the five subsets (20% of the entire dataset) taking turns serving as the test set. When one subset acts as the test set, the remaining four subsets (80% of the data) constitute the training set. In the second stage, a similar random partition is applied to the training set, acting as a cross-validation method. During the cross-validation, we perform a grid search over ranges of different hyperparameters for RR and RFR. For RR, the strength of L2-regularization (i.e., alpha) is set

to range from 1 to 399, with an incremental step of 2. For RFR, the hyperparameters used in this study is consistent with those of Weng et al. (2022). Finally, the modelled values of the test sets are used to reflect meteorologically driven ozone variabilities. Trends estimated from the predicted ozone are therefore indicative of meteorologically driven ozone trends, while the residuals between observed and predicted values (observed minus predicted values) can reflect emission-driven ozone trends (Li et al., 2019b). We conduct the above analyses for the summertime period of 2013-2019 to quantify the attribution of surface ozone

trends in China. The model performance and interpretations will be discussed in Section 5.2.

### 2.3 Chemical transport model

In this study, we employ four simulations generated from three chemical transport models, two on a global scale (GEOS-Chem with coarse resolution and CAM4-chem) and two on a regional scale (GEOS-Chem with fine resolution and WRF-CMAQ), to quantify the impact of emission changes and meteorology on the trend of tropospheric ozone over ESEA. Sections 2.3.1 to

2.3.3 describe the configurations of the three models, Section 2.3.4 compares the key differences in capability and configuration among the models.

### 2.3.1 GEOS-Chem

GEOS-Chem is a state-of-art global to regional three-dimensional chemical transport model (Bey et al., 2001). We apply GEOS-Chem version 13.3.1 (available at https://github.com/geoschem/GCClassic/tree/13.3.1, last accessed: 23th July, 2024).

The model is driven by MERRA2 re-analysis meteorological fields. In short, GEOS-Chem describes a comprehensive stratosphere-troposphere coupled ozone–$NO_x$–VOCs–aerosol–halogen chemistry scheme (Eastham et al., 2014), and includes



online calculation of dry and wet depositions of gases and aerosols. More detailed descriptions of the GEOS-Chem chemistry, transport and mixing scheme, and deposition are provided by Wang et al. (2022a).

Global anthropogenic emissions used in our GEOS-Chem simulations are from Community Emissions Data System inventory (CEDSv2), which builds on the extension of the CEDS system to 2017 as described in McDuffie et al. (2020) (O'Rourke et al., 2021). Emission estimates in the CEDSv2 inventory (Fig.2) are scaled to existing authoritative inventories as a function of emission sector and fuel type where available. In Asia, the authoritative emission inventory employed for this scaling procedure includes the Regional Emission inventory in Asia (REAS) over Asia (Kurokawa et al., 2013), the Multi-resolution Emission

Inventory model for Climate and air pollution research (MEIC) over China (Zheng et al., 2018), NIER inventory over South Korea (South Korea National Institute of Environmental Research (2016)), and the SMoG-India inventory (Venkataraman et al., 2018) over India. We also include yearly global aircraft emissions from the CEDSv2 inventory to account for their impacts on tropospheric chemistry following Wang et al. (2022a). For natural emissions, GEOS-Chem includes online calculation of biogenic VOCs emissions (Guenther et al., 2012), and $NO_x$ emissions from soil (Hudman et al., 2012; Lu et al., 2021a) and

lightning (Murray et al., 2013). Biomass burning emissions are from the BB4CMIP inventory (van Marle et al., 2017), in which the emissions for years after 1997 are identical to the Global Fire Emissions Database version 4 (GFED4; van der Werf et al. (2017)). Surface methane concentration in GEOS-Chem is prescribed based on spatially interpolated monthly mean surface methane observations from the NOAA Global Monitoring Division, while the transport and chemistry of methane is simulated interactively. The use of methane boundary conditions instead of methane emissions is to ensure a realistic methane

distribution in the model, as there are significant uncertainties associated with bottom-up methane emission inventories (Lu et al., 2021b).

We apply the GEOS-Chem model to simulate tropospheric ozone change from 1995 to 2019 on both global and regional scales. For the global scale, we run the model at a horizontal resolution of 4°× 5°, with 72 vertical layers extending from the surface

to 0.01 hPa. The three-hourly global concentrations of atmospheric compositions are then archived as boundary conditions to drive the nested model simulation over the nested East and Southeast Asia domain (60°-150°E, 11°S-55°N) at the horizontal resolution of 0.5°× 0.625°. Three-dimensional ozone concentrations are output hourly to allow the co-sampling with IAGOS and ozonesonde measurement.

The simulation strategy for quantifying the attribution of ozone trends mostly follows Wang et al. (2022a) (Table 2). We conduct the standard simulation (BASE) from 1995 to 2019 using year-specific meteorology fields and emissions as described above. In the Fix_Globe_AC simulation, we fix global anthropogenic emissions (including aircraft emissions) and methane concentration at their 1995 levels. As a result, ozone changes in the Fix_Globe_AC simulation only reflect variations in natural emissions and climate conditions, so it estimates the climatic influence on tropospheric ozone trends. The difference in ozone

trends between the BASE and Fix_Globe_AC simulation can be used to quantify the contribution of global anthropogenic



emissions of tropospheric ozone precursors to ozone trends. We also conduct a simulation Fix_ESA_A by fixing anthropogenic emissions over ESEA countries (Fig.1a) at their 1995 levels. We use the initial chemical fields archived in Wang et al. (2022a) to drive the model simulation from 1995, in which the initial chemical fields have been spun-up for ten-year to ensure the adequate distributions of chemical species in the stratosphere. For global simulations at a 4°× 5° resolution, our simulation

spans the entire year from 1995 to 2019. For regional simulations at a 0.5°× 0.625° resolution, we constrain our simulations to boreal summer (June, July, August, with simulation in May as spin-ups) and conduct simulations for the years 1995, 2000, 2005, 2010, 2013, 2015, 2017, and 2019.

### 2.3.2 CAM4-chem

We perform two 19-year-long simulations (2000-2018) with the Community Atmosphere Model version 4 with chemistry

(CAM4-chem), a component of the Community Earth System Model version 1.2.2 (CESM; Lamarque et al., 2012) aided with the TOAST ozone tagging technique as described in Butler et al. (2018, 2020). The two simulations attribute the simulated ozone in terms of its $NO_x$ and reactive carbon (RC, NMVOC+CO+$CH_4$) sources respectively. A one-year spin-up was performed for the $NO_x$-tagged simulations and a two-year spin-up for the RC-tagged simulations.

Anthropogenic emissions of $NO_x$, CO, non-methane volatile organic compounds (NMVOCs), $NH_3$, $SO_2$, and PM are taken

from the recently launched Hemispheric Transport of Air Pollution version 3 (HTAPv3; Crippa et al., 2023) emissions inventory. We specify aircraft emissions at three sets of altitude ranges representing the different flight phases (landing/take-off, ascent/descent, and cruising). Biomass burning emissions are from GFED-v4 inventory (van der Werf et al., 2010). The biogenic NMVOC emissions are from CAMS-GLOB-BIO-v3.0 (Sindelarova et al., 2021), and biogenic $NO_x$ (from soil) is prescribed as in Tilmes et al., (2015). Same as GEOS-Chem, methane concentration is imposed as a surface boundary condition.


The chemical mechanism applies the MOZART-4 tropospheric chemical mechanism (Emmons et al., 2010), which is modified to include tagged ozone tracers as described in Butler et al. (2018). The mechanism contains detailed chemistry of methane and NMVOC oxidation, but does not contain any halogen species. Stratospheric ozone is formed through photolysis of molecular oxygen, and is fixed at the upper model boundary based on output from CESM2-WACCM6 (Emmons et al., 2020).


Separate tag identities are specified for regional land-based emissions and for global biogenic, biomass burning, aircraft and shipping emissions of ozone precursors, as well as for ozone from production in the stratosphere (Nalam et al., 2024). A total of 13 regions are tagged for $NO_x$ emissions, including East Asia and Southeast Asia (Fig. S2). The $NO_x$-tagged simulations also contain a tag for ozone produced from lightning $NO_x$, and the RC-tagged simulations contain an additional tag for ozone

produced from methane oxidation. In both $NO_x$- and RC-tagged simulations, the sum of tagged ozone contributions is equal to the total ozone simulated by the model.



The model is run at a horizontal resolution of 1.9°×2.5°, with 56 vertical levels (from surface to 1.86hPa) for the 2000-2018 period driven by meteorological data from the MERRA2 reanalysis. The temperature, horizontal winds, and surface fluxes

from MERRA2 reanalysis dataset are nudged every time step (30 minutes) by 10 % towards analysis fields.

### 2.3.3 WRF-CMAQ

We also apply the Community Multiscale Air Quality (CMAQ) (version 5.2.1) three-dimensional regional air quality model. Meteorological input of the CMAQ model is provided by the Weather Research and Forecasting (WRF v3.9) model. The initial and boundary conditions of meteorological fields are generated from the ERA5 reanalysis dataset for years before 2000, and

from the National Center for Environmental Prediction (NCEP) FNL Operational Model Global Tropospheric Analyses with a horizontal resolution of 1° × 1° for years after 2000.

The physical schemes used in the WRF simulation is summarized in Table S2. We use SAPRC07TIC (Carter, 2010; Hutzell et al., 2012) for gas-phase chemistry and AERO6i (Murphy et al., 2017; Pye et al., 2017) for aerosols. However, the model

does not include specified stratospheric chemistry. Anthropogenic emissions are derived from the MIX inventory (Li et al., 2017) as of the 2010 level, with interannual variations being scaled in accordance with the CEDSv2 inventory. Biomass burning emissions are identical to those used in GEOS-Chem model. In addition, we also add hourly soil emissions of $NO_x$ calculated online from the GEOS-Chem model to the CMAQ simulations as offline emissions. The biogenic emissions are calculated using the Model of Emissions of Gases and Aerosols from Nature (MEGAN) (Guenther et al., 2012) driven by the

meteorological outputs from the WRF model. However, the model does not consider emissions in the upper troposphere, such as lightning emissions and aircraft emissions. Methane concentration is a fixed value of 1850 ppbv in the simulation.

The WRF-CMAQ model domain covers ESEA at a horizontal resolution of 36 km × 36 km, as shown in Figure S3. We set 23 vertical layers extending from the surface to the height of ~22 km. In particular, the chemical boundary conditions are generated

from the GEOS-Chem simulation using the newly developed GC2CMAQ tool (Zhu et al., 2023). As such, the boundary conditions used in the GEOS-Chem nested model and CMAQ model are largely reconciled.

We conduct WRF-CMAQ simulations for the months of June, July, and August for the years 1995, 2000, 2005, 2010, 2013, 2015, 2017, and 2019. These simulation years align with the GEOS-Chem simulations at nested grid. The initial 11 days before

each June are considered as the spin-up time for the simulations. We also perform four sensitivity simulations following the same strategy as the GEOS-Chem simulations.

### 2.3.4 Comparison of GEOS-Chem, CAM4-chem, and WRF-CMAQ model characteristics and configuration

One of the main purposes for employing three chemical models with distinct model characteristics is to assess the consistency and discrepancies among these models in reproducing current levels and trends in tropospheric ozone across East Asia, using



the same observational dataset as a benchmark. Additionally, it allows us to quantify the uncertainty of ozone trend attributions, considering variations in the model's spatial resolution, meteorological input, and emission data. Table 3 compares the key differences between GEOS-Chem, CAM4-chem, and WRF-CMAQ simulation used in this study.

First, the three models use different meteorological fields. GEOS-Chem and CAM4-chem do not simulate meteorological
fields; rather, they use MERRA2 re-analysis data assimilated from multiple observations. CMAQ model uses WRF simulated meteorological fields as input. All models are offline and do not account for the interactions between atmospheric chemistry and meteorology.

Second, the chemical schemes among the three models are largely different. The GEOS-Chem version 13.3.1 describes a
stratosphere-troposphere coupled ozone–$NO_x$–VOCs–aerosol–halogen chemistry scheme. In particular, the model includes a detailed halogen chemistry that tends to provide additional ozone chemical loss especially in the free troposphere. CAM4-chem applies the MOZART-4 tropospheric chemical mechanism but does not contain halogen species. Stratospheric chemistry is also simplified by only considering the ozone formed through photolysis of molecular oxygen, with a fixed upper boundary condition for ozone as described in Nalam et al. (2024). CMAQ used SAPRC07TIC for gas-phase chemistry and AERO6i for
aerosol mechanism. It does not consider stratospheric chemistry. Both GEOS-Chem and CAM4-chem consider the interannual variation of methane concentrations, while the CMAQ model treats methane concentration as a fixed level for all years.

Third, the three models do not share the same emission input. For anthropogenic emissions, GEOS-Chem model applies the CEDSv2, CAM4-chem applies the HTAPv3 inventory, while CMAQ applies MIX inventory with interannual variations being
scaled by the CEDSv2 inventory. Additionally, the models differ in their consideration of natural emissions. All the models incorporate biogenic VOCs emissions from the MEGAN algorithm; however, due to variations in meteorological fields, the emission amounts are expected to differ. The GEOS-Chem and CMAQ models utilize the same soil $NO_x$ emission inventory, whereas the CAM4-chem model employs a different approach for estimating soil $NO_x$ emissions. The CMAQ model does not account for lightning emissions and aircraft emissions.

Fourth, the horizontal and vertical resolutions differ among the three models. GEOS-Chem utilizes two horizontal resolutions in this study of 4°×5° and 0.5°×0.625°; CAM4-chem operates at 1.9°×2.5°; and WRF-CMAQ employs the finest resolution of 36 km × 36 km. In terms of vertical resolution, GEOS-Chem comprises 72 layers extending from the surface to 0.01 hPa, CAM4-chem includes 56 vertical levels, and WRF-CMAQ has the coarsest vertical resolution with 23 layers extending to 50
hPa.



## 2.4 Trend estimation

### 2.4.1 Generalized least-squares method for surface measurements

For surface measurement, we derive the parametric linear ozone trend at each monitoring site using the generalized least-squares method. As ozone has a strong seasonal cycle, estimating trends based on monthly mean anomalies is more accurate than using the monthly mean data, if there were missing data (Cooper et al., 2020; Lu et al., 2020). We first derive the monthly mean anomalies of ozone by subtracting the original values from the monthly mean data. We then estimate linear trends using the generalized least-squares method, and report the linear trend coefficient and corresponding p-values.

### 2.4.2 Quantile trend estimation for IAGOS and ozonesonde observations

For IAGOS and ozonesonde observations, we use the quantile regression method (Koenker and Bassett, 1978) to derive tropospheric ozone trends following the methodology outlined by Gaudel et al. (2020). This method is advantageous for trend estimates for time series with intermittent missing values and temporal discontinuities, as it relies on the rank value of the sample distributions rather than mean value (Koenker and Xiao, 2002; Chang et al., 2021). We aaply the same procedures, such as deseasonalization, detailed in Section 2.4 of Wang et al. (2022a). Linear trends (in ppbv decade$^{-1}$) of ozone at the 50$^{th}$ percentile  (median) for the period 1995–2019 are reported with a corresponding p-value.

## 3 Trends in emissions and meteorological variables over East and Southeast Asia

### 3.1 Trends in anthropogenic emissions of ozone precursors

Figure 2 shows the spatial distribution (averaged over 2015, 2017, and 2019, in accordance with the model simulation years) and trends (1995-2019) in summertime anthropogenic $NO_x$, CO, and NMVOCs emissions derived from the CEDSv2 inventory. The spatial distributions of these emissions of ozone precursors are similar, with high emissions concentrated in the populated regions over the ESEA region, including the North China Plain (NCP), Yangtze River Delta (YRD), Pearl River Delta (PRD), the Sichuan Basin (SCB), South Korea, Japan, southern Thailand, southern Vietnam, and central Indonesia.

Figures 2b and 2c illustrate the emission trends from 1995 to 2019. Anthropogenic emissions of $NO_x$, CO, and NMVOCs have increased by 129%, 17%, and 50% from 1995 to 2019, averaged over the continental ESEA, respectively. $NO_x$ emissions in China, Southeast Asia, Indonesia, and India have increased during this period, contrasting with a decline in relatively developed countries such as South Korea and Japan. In China, $NO_x$ emissions surged by 4.7 times from 1995 to 2011, followed by a 29% reduction from 2011 to 2019 due to the implementation of stringent emission control measures. Emissions in Indonesia and Southeast Asia have grown by 3.7 and 2.1 times over these 25 years, respectively, while South Korea and Japan have reduced their $NO_x$ emissions by 20% and 60% since 1995, respectively.





CO emissions show similar trends as NO$_x$. In China, CO emissions increased by 80% from 1995 to 2008, followed by a 33% decrease by 2019. Emissions in Southeast Asia continued to rise by approximately 50% from 1995 to 2019. In Indonesia, emissions peaked with a 67% increase in 2013 relative to 1995 level, followed by a 15% reduction by 2019. In South Korea and Japan, CO emissions decreased by 45% and 56% in 2019 compared to 1995, respectively.


NMVOC emissions show a different trend. Most countries in the ESEA region have experienced an increase in NMVOC emissions, with an exception of Japan, where emissions decreased by approximately 40% from 1995 to 2011 and have remained stable. NMVOC emissions in China, South Korea, and Southeast Asia have increased by 51%, 47%, and 67% from 1995 to 2019 respectively. However, emissions in these three countries have changed little in the last decade.

**3.2 Trends in meteorological variables and natural emissions**

Figure S4 shows the trends in key meteorological parameters relevant to ozone natural sources, chemistry, and transport from 1995 to 2019 over ESEA, derived from the MERRA2 reanalysis dataset. A notable upward trend in surface downward solar radiation is discernible across Southeast Asia, whereas a decline is evident in most parts of China. These shifts in solar radiation align with trends in total cloud coverage. In terms of temperature, the ESEA region has witnessed widespread warming, with 400 an exception of decreasing temperatures in Myanmar. Specific humidity exhibits an increasing trend across most of the ESEA region, with largest increases observed in China and Myanmar. Trends in surface wind speed vary across regions. The eastern China and South Korea has experienced a decrease in wind speed, which may be attributed to rapid urbanization in these areas. In contrast, wind speed over the South China Sea shows increasing trend.

Figure S5 displays the spatial distribution of summertime mean emissions of biogenic volatile organic compounds (BVOCs), soil NO, lightning NO, and biomass burning CO across ESEA, averaged over the years 2015, 2017, and 2019. Emissions from vegetation, soil, and lightning are calculated by the parameterization schemes implemented in GEOS-Chem driven by MERRA2 meteorological data, while biomass burning emissions are derived from the inventory (Section 2.3.1). BVOC emissions are high in the southern China, Southeast Asia, and central Indonesia, where vegetated and forested areas are most 410 prominent. Soil NO emissions are high in regions with intensive agricultural fertilizer application and nitrogen deposition, such as the NCP in China and Indo-Gangetic Plain in India (Lu et al., 2021a). High lightning NO emissions are concentrated over northern China and regions near Mongolia, reflecting a larger amount of NO released per flash (500 moles) for the lightning north of 35◦ N in Eurasia and compared to 260 moles for other regions used in the parameterization scheme (Lu et al., 2019b). Biomass burning emissions are the most intensive in Indonesia during boreal summer.


Figure S5 also illustrates the temporal trends in these natural emissions. Significant positive trends in BVOC emissions are shown in eastern China, Southeast Asia, and parts of India, in contrast to a significant decline in Myanmar. This is mostly likely driven by rising temperatures (Figure S4b). Soil NO emissions show large interannual variability. Here we do not





consider changes in fertilizer applications so that trends are mainly driven by meteorological conditions such as soil moisture
and temperature. Increases in lightning NO emissions in northern China and Mongolia are likely linked to the intensification
of thunderstorm activity. Biomass burning emissions show substantial interannual variability. For example, in the year 1997,
biomass burning emissions are ~100 times higher than the year 1995. These trends are expected to influence the ozone trend
and variability on the top of anthropogenic emission-driven trends.

## 4 Evaluation of the capability of chemical transport models in reproducing ozone and trends

**4.1 Present day level of ozone vertical profile and surface concentration**

Figure 3 compares the simulated summertime tropospheric ozone profiles with observations from IAGOS and ozonesonde
measurements for the present-day period (2015-2019). We sample hourly three-dimensional ozone concentrations from the
GEOS-Chem (at both 4°×5° and 0.5°×0.625° resolution) and the WRF-CMAQ model along the IAGOS flight tracks and at
ozonesonde sample time for the years 2015, 2017, and 2019. For comparison with IAGOS profiles, we average the all profiles
for East Asia (103°-137°E, 27-46°N) and Southeast Asia (90°-123°E, -10°-27°N) domains. For comparison with ozonesonde
observations, we average ozone profiles at sites of Pohang, Sapporo, Tateno (Tsukuba), and Naha to represent East Asia, and
King's Park, Hanoi, and Kuala Lumpur to represent Southeast Asia. As hourly output of three-dimensional ozone
concentrations from the CAM4-chem model is not available, we indirectly evaluate its performance by comparing vertical
ozone distributions averaged over East Asia and Southeast Asia domains in CAM4-chem to GEOS-Chem simulation at
0.5°×0.625° resolution.

Observations from the IAGOS database show an ozone peak near the 900 hPa level in both East Asia and Southeast Asia,
indicative of elevated ozone concentrations in the boundary layer above densely populated areas with high anthropogenic
activities. Above the 900 hPa level, ozone concentrations initially decrease and then increase with altitude. Southeast Asia
exhibits a less pronounced vertical gradient in ozone concentrations compared to East Asia, reflecting the more convective
environment prevalent in tropical regions which facilitates vertical transport and mixing of ozone in the troposphere. In
addition, the more active stratosphere-troposphere ozone transport in the midlatitudes also contributes to the larger ozone
vertical gradient in East Asia. Similar vertical ozone structures are evident in ozonesonde observations.

We find that, overall, all models applied in this study capture the observed ozone vertical profiles over East Asia and Southeast
Asia. The GEOS-Chem model at fine (0.5°×0.625°) resolution (hereafter referred to as GC05) effectively replicates the ozone
peak observed in the IAGOS profiles at near the 900 hPa level above both East Asia and Southeast Asia, with a small bias of
6-8 ppbv, respectively. It shows no prominent ozone bias when compared to the IAGOS profiles in the middle and upper
troposphere in East Asia, but it underestimates ozone concentrations by 10-15 ppbv in the upper troposphere over Southeast
Asia. It also reproduces the ozone vertical structure observed from ozonesonde measurement, yet it displays a high bias in the



lower troposphere of 10-20 ppbv and a low bias of 15 ppbv in the upper troposphere across both East Asia and Southeast Asia. We also find that GEOS-Chem simulations at both coarse (4°×5°) and fine (0.5°×0.625°) resolutions show no significant discrepancies in ozone concentrations in the free troposphere. This can be attributed to the sufficiently long chemical lifetime of ozone in the free troposphere as such ozone is relatively well mixed (Petetin et al., 2018; Wang et al., 2022a). The WRF-
CMAQ model shows comparable ability to capture the observed ozone vertical structure in East Asia and Southeast Asia, but shows excessive high bias in Southeast Asia and in the upper troposphere and lower stratosphere, due to the lack of a detailed description of stratospheric chemistry. The CAM4-chem model results are mostly consistent with the GC05 model in simulating ozone vertical profiles across both East Asia and Southeast Asia.

Figure 4 evaluates the simulated summertime surface MDA8 ozone concentrations across ESEA for 2017, when observations in all regions and output from all three models are available. Observations indicate high summertime MDA8 ozone concentrations in the NCP (73±21 ppbv), YRD (59±15 ppbv), SCB (56±14 ppbv) in China, reflecting intensive emissions of ozone precursors and active photochemistry in these populous city clusters, followed by South Korea (48±15 ppbv) and Japan (45±10 ppbv). In comparison, the PRD region in China, Thailand, and Malaysia show relatively low summertime MDA8
ozone of 37±17, 25±3 and 31±8 ppbv due to the effect of the summer monsoon (Zhou et al., 2013; Lu et al, 2018a; Gao et al., 2020b). Ozone concentrations in the PRD region typically peak in boreal autumn.

As shown in Figure 4, The WRF-CMAQ model shows a relatively good agreement with observed MDA8 ozone levels, exhibiting a moderate overestimation of 3-8 ppbv in the NCP, YRD, and Japan, and Thailand. However, the overestimation is
more pronounced over the PRD (10 ppbv) and Malaysia (20 ppbv). The CAM4-chem model well captures surface ozone concentrations in the NCP and SCB in China, and Japan, while shows a slight high bias of 8-10 ppbv in the PRD region and South Korea. In comparison, the GC05 model demonstrates a substantial overestimation of 9-20 ppbv across all examined regions.

Overall, all models capture the spatial distributions of surface ozone over ESEA, as indicated by the high spatial correlation coefficients between the observed and simulated values ranging from 0.50-0.78 (except for Thailand where only 11 sites are available), but they tend to overestimate surface ozone concentrations over ESEA. This overestimation highlights a recurring challenge for models operating at relatively coarse resolutions (30 km or coarser) in accurately representing surface ozone levels in densely populated regions, characterized by intense anthropogenic emissions and rapid chemical conversion. Such
high bias reflects a complex combination of multiple factors (Li J. et al., 2019c; Yang and Zhao, 2023). A model grid with a horizontal resolution of 30 km or coarser may not resolve the heterogeneity of anthropogenic emissions and thus lead to artificial mixing of ozone precursors, causing either higher or lower ozone production efficiency and ozone biases (Yu et al., 2016; Young et al, 2018). In addition, with coarser model resolution, representative issues emerge when comparing gridded simulated results to site-level observation, and the model has increasing difficulty to presenting local meteorological conditions



particularly over complex terrain. Yang and Zhao (2023) provided clear evidence that correlation coefficients between simulated and observed ozone concentrations in China decrease with decreasing horizontal resolution in air quality models. Here, we also find smaller model-to-observation bias from the same GEOS-Chem model configuration but at 0.5°×0.625° compared to that at 4°×5° resolution (results not shown). However, conducting fine (e.g., 10 km or higher) resolution chemical transport models at a large spatial domain (such as ESEA) significantly enhances the computational costs.

Our GEOS-Chem simulation configured for this study (using version 13.3.1 and CEDSv2 as anthropogenic emission inventory) shows particularly prominent high summertime ozone bias at city clusters in China. This high bias is not found or at least not prominent in previous studies using earlier GEOS-Chem model version (e.g., version 11) and the MEIC inventory for anthropogenic emissions (Lu et al., 2019b; Li et al., 2019b,c; Tan et al., 2023). A possible reason for this discrepancy could be the integration of updated aromatic chemistry in GEOS-Chem models from version 13.0.0 onwards. This update has been

shown to elevate surface ozone concentrations by at least 5 ppbv in eastern China (Bates et al., 2021). The use of the CEDSv2 inventory in GEOS-Chem intends to standardize emissions inventories across all countries. However, it might be less accurate for simulating air pollution over China compared to the MEIC inventory, which employs more localized data for activity levels and emission factors (Zheng et al., 2018). Although it is widely recognized that uncertainties in emission inventories and meteorological fields contribute to simulated ozone biases, conducting sensitivity simulations with a broader array of emission

inventories and meteorological fields to pinpoint and minimize these uncertainties would entail substantially higher costs and is beyond the scope of this study.

**4.2 1995-2019 ozone trends in the troposphere and at the surface**

We proceed to examine the capability of the models in reproducing the long-term summertime ozone trends in ESEA from 1995 to 2019. Figure 5 presents the observed ozone trends at the 50th percentiles at each vertical layers (from 950 to 200 hPa

at 50 hPa intervals) based on IAGOS and ozonesonde measurements, estimated by the quantile regression model as described in Section 2.4. Both IAGOS and ozonesonde observations indicate increasing tropospheric ozone over ESEA since 1995, consistent with previous studies (Gaudel et al., 2020; Wang et al., 2022a). However, the structure of ozone trends differs between the IAGOS and ozonesonde measurement, reflecting the difference in the sampling regions and time. For IAGOS profiles in East Asia, the rate of ozone increase reaches 8 ppbv decade$^{-1}$ below the 900 hPa level. The rate of increase decreases

initially with altitude, but rises again in the upper troposphere. For ozonesonde observations, the ozone increasing rate rises from 4 ppbv decade$^{-1}$ below 900 hPa to 10 ppbv decade$^{-1}$ in the upper troposphere. In Southeast Asia, increasing ozone trends are evident along the IAGOS profiles, reaching 10 ppbv decade$^{-1}$ across the troposphere. In comparison, trends measured at the 3 ozonesonde sites (King's Park, Hanoi, and Kuala Lumpur) are with large uncertainty, highlighting the challenges in ozone trend estimate with limited samples (Chang et al., 2022).


Since only GEOS-Chem is applied for the continuous 1995-2019 simulation with full three-dimensional hourly output of ozone concentrations, we rely on GEOS-Chem simulation for direct comparison with IAGOS and ozonesonde observations (Fig. 5),





and use the GEOS-Chem result as an intermediary platform to indirectly evaluate the overall ozone variation since 1995 from the CAM4-chem and CMAQ models (Fig. S6). We find that GC05 mostly reproduces the notable tropospheric ozone increase

in ESEA and also the different structure measured from the IAGOS and ozonesonde profiles. In East Asia, it aligns closely with the observed ozone trends for the IAGOS profiles, but underestimates the rate of ozone increase at ozonesonde sites. In Southeast Asia, although GC05 model does simulate the overall ozone increase in the troposphere from 1995 to 2019 (Fig.5), it underestimates the rate in the lower troposphere compared to both IAGOS and ozonesonde profiles. In general, we find that the GC05 outperforms GC45 in reproducing tropospheric ozone increases in ESEA, except for the lower troposphere over

Southeast Asia.

Figure S6 compares simulated tropospheric ozone trends averaged over East Asia and Southeast Asia from GEOS-Chem, CAM4-chem, and WRF-CMAQ. All models concur on the notable tropospheric ozone increases in the period of 1995-2019. Even though GEOS-Chem underestimates ozone trends measured in the IAGOS and ozonesonde profiles (Fig.5), it simulates

the largest ozone increases compared to the CAM4-chem and WRF-CMAQ model results. These results highlight a common difficulty in chemical models to capture long-term tropospheric ozone trends in ESEA, especially Southeast Asia (Wang et al., 2022a, Wang et al., 2022b). Wang et al. (2022b) shows that constraining $NO_x$ emissions from satellite observations can improve GEOS-Chem's ability to reproduce the observed ozone trends over the Peninsular Southeast Asia during 2005-2016, indicating that $NO_x$ emission growth may have been underestimated in the current emission inventory. Shah et al. (2024) shows

that increasing nitrate photolysis in the free troposphere could substantially addresses the underestimation of tropospheric ozone trends in chemical models.

Figure 6 presents the observed and simulated mean summertime surface ozone variations at available monitoring sites from 1995-2019. For Japan and South Korea, we analyze ozone trends from 1995 (295 sites with at least 20 years with available

observations in 1995-2019) and 2001 (87 sites with at least 16 years with available observations in 2001-2019) from their respective national monitoring network. For Thailand and Hong Kong China, 6 and 9 long-term monitoring sites are available. Nationwide ozone monitoring network in mainland China was not available before 2013. We apply observations at five individual stations with different time span (Xu et al., 2020) (Table1). The Chinese Meteorology Administration (CMA) and Gucheng (GCH) sites are in close proximity and share similar ozone variation characteristics, therefore they are grouped.

Simulated ozone concentrations at corresponding time and model grid at individual monitoring sites from the GC05, CAM4-chem, and WRF-CMAQ are used to compare with the observations. We focus on the evaluation of long-term trends.

Observed summertime surface ozone concentrations have increased at 2.4 ppbv decade[-1] (p-value ≤ 0.1) in Japan from 1995 to 2019 and at 6.9 ppbv decade[-1] (p-value ≤ 0.1) in South Korea from 2001 to 2019 (Figure 6). The GC05 model captures 55.0%

(3.8 ppbv decade[-1]) of the observed ozone increases in South Korea but only shows a small rate of increase at 0.60 ppbv decade[-1] (25% of the observed trend) in Japan. Thailand and Hong Kong, China exhibit surface ozone increases of 2.4 and 3.7 ppbv



decade$^{-1}$ (p-value ≤ 0.1), respectively, and these trends are also captured by the GC05 model (46% for Thailand and 81% for Hong Kong, China). The CAM4-chem results are available for 2000-2018 and show similar underestimation of surface ozone trends. Both models largely reproduce the observed interannual variability in surface ozone concentrations in Japan, South

Korea, Thailand, and Hong Kong, China, as indicated by temporal correlation coefficients ranging from 0.59 to 0.84 for GC05 and from 0.45 to 0.85 for CAM4-chem. The WRF-CMAQ model is only run for seven years during the 1995-2019 period, which is insufficient for deriving long-term trends. As shown in Figure 6, it still simulates an increase in surface ozone concentrations in these regions, although it similarly underestimates the trends.

Observed trends in summertime surface ozone concentrations are not consistent among the five sites in mainland China. Ozone concentrations show significant increases in northeastern China (LFS site) and in the NCP region (SDZ, CMA and GCH sites). Both GC05 and CAM4-chem models reproduce these ozone increases, but significantly underestimate the trends at the SDZ and LFS sites (1.7-5.3 ppbv decade$^{-1}$ in the simulations versus 9.7-12.5 ppbv decade$^{-1}$ in the observations), while they match the trends at the CMA ad GCH sites (6.7-7.4 ppbv decade$^{-1}$ in the simulations versus 6.0 ppbv decade$^{-1}$ in the observation). At

the LA site in eastern China, observations indicate a slight increase (0.2 ppbv decade$^{-1}$), whereas both CAM4-chem and GC05 models simulate notable increase of 3.3-4.4 ppbv decade$^{-1}$, although they do capture the interannual variability. Overall, all models indicate long-term increases in summertime mean surface ozone concentrations since 1995 or the early 2000s at most sites across ESEA, although the magnitudes are biased compared to the observations.

## 5 Attribution of tropospheric ozone trends over East and Southeast Asia

### 5.1 1995-2019 trends

#### 5.1.1 Tropospheric (950-200hPa) ozone

We quantify the factors contributing to summertime tropospheric ozone changes in ESEA from 1995 to 2019 using model sensitivity simulations. Figure 7 displays the spatial difference in tropospheric (950-200 hPa) column ozone mixing ratio (TCO) in 2005, 2013, and 2019 relative to the 1995 level from GEOS-Chem (hereafter referred to as GEOS-Chem at 0.5° resolution)

and the WRF-CMAQ model. Figure 8 summarizes the impact of anthropogenic emissions and climate change on the variation of tropospheric ozone burden over ESEA (including the East Asia domain of 80°E-145°E, 30°N-53°N and Southeast Asia domain of 92.5°E-135°E, 10°S-30°N). Our base simulation from GEOS-Chem indicates a tropospheric ozone burden of 24 Tg over the ESEA during JJA 2019, representing a 16% increase from 1995.

Both GEOS-Chem and WRF-CMAQ agree that change in anthropogenic emissions is a key driver of the tropospheric ozone increase over ESEA from 1995 to 2019. GEOS-Chem (WRF-CMAQ) quantifies that shifts in global anthropogenic emissions enhance TCO averaged across continental ESEA (ocean area excluded) by 2.3 (2.9), 4.2 (5.2), 4.2 (5.7) ppbv in 2005, 2013,



and 2019, respectively, relative to the 1995 level (Fig. 7b). These increases account for 47% and 71% respectively of the simulated TCO enhancement from the GEOS-Chem and WRF-CMAQ model between 1995 and 2019. In terms of the tropospheric ozone burden, GEOS-Chem and WRF-CMAQ estimate that 53% and 59% of the increase over ESEA from 1995 to 2019 can be attributed to increasing global anthropogenic emissions, respectively (Fig. 8).

A comparison of Figures 7c and d reveals that the increase in emission-driven tropospheric ozone over ESEA is primarily attributed to changes in emissions within, rather than outside, ESEA. GEOS-Chem estimates that the increase in TCO driven by emission changes inside ESEA is 2.8 ppbv (66% of the emission-driven TCO change), compared to those outside ESEA of 1.5 ppbv, in 2019 relative to 1995 level. The CMAQ model simulates a more pronounced partitioning of the ozone increase towards emissions within ESEA (4.4 ppbv, 77% of the emission-driven TCO change) versus those outside (1.3 ppbv). The TCO increase contributed by emissions within ESEA also exhibits notable spatiotemporal variations. From 1995 to 2013, anthropogenic emissions of ozone precursors in ESEA increase steadily (Fig.2), leading to notable TCO enhancement in eastern and central China (6-10 ppbv from GEOS-Chem and 12-15 ppbv in CMAQ), the Korean Peninsula (3-6 ppbv from both models), and in Indonesia (2-5 ppbv). However, the emission-driven TCO increase decelerates or even reduces thereafter (Fig.7c), coinciding with the reduction in anthropogenic emissions of $NO_x$ and CO from 2013 (Fig.2). This reduction in emission largely reflects the enactment of the Action Plan on Air Pollution Prevention and Control in China initiated in 2013. Figure 8 further illustrates that ESEA emissions have slightly reduced the tropospheric ozone burden after 2013 by 0.2 Tg, suggesting that efforts to mitigate air pollution in China have slowed or even halted the tropospheric ozone rise in ESEA.

Figure 7d illustrates the continuous rise in ozone enhancement due to emissions originating outside of ESEA from 1995 to 2019. It also shows distinct spatial distributions that are significantly different from those driven by emissions within ESEA (Fig.7c). Both GEOS-Chem and CMAQ models simulate an ozone enhancement attributable to emissions outside ESEA ranging from 3-5 ppbv in Tibet China, and Indochina relative to the 1995 level, with smaller enhancements in other regions. We also find that changes in impact of emission changes outside of ESEA becomes increasingly important at higher altitudes (Fig.9), consistent with the findings from previous studies (Ni et al., 2018). With the overall decline in anthropogenic emissions in Europe and North America, South Asia has emerged as a key region influencing the tropospheric ozone trend in ESEA, as also indicated by Figure 7. Prior studies have documented significant increases in tropospheric ozone since the 1990s over India, propelled by rising anthropogenic emissions of pollutants (Lu et al., 2018a; Wang et al., 2022a). The South Asian monsoon is anticipated to transport ozone and its precursors from South Asia to the downwind ESEA region, thereby contributing to the ozone enhancement there.

The tagged ozone module in CAM4-chem offers an independent assessment of the regional anthropogenic contribution to ozone concentration in East Asia and Southeast Asia (Fig.10 for ozone produced from $NO_x$ emissions, Fig.S7 for reactive carbon emissions). We note here that the definition of region (Fig.S3) in the CAM4-chem tagged simulation is not consistent



with the region defined in this study. Focusing on East Asia, we find that tropospheric ozone produced from anthropogenic $NO_x$ emissions within East Asia increase by 4 ppbv from 2000 to 2012, acting as the main region contributing to anthropogenic ozone enhancement in this period (Fig.10a). The ozone enhancement then decreases after 2013, consistent with the GEOS-Chem and CMAQ model estimation and again reflects the emission change in China. Emissions outside East Asia also contribute to tropospheric ozone increases. The combined ozone produced by anthropogenic $NO_x$ emissions in Southeast Aisa and South Asia contributed to a rise of 3 ppbv in TCO over East Asia from 2000 to 2018, indicating increasing import of ozone pollution from these regions to East Asia. In contrast, contributions from Europe and North America decreased by 2-3 ppbv from 2000 to 2018. This is an expected result of anthropogenic emission controls of ozone precursors in Europe and North America. Results from the RC-tagged simulation (Fig.S7) show similar patterns. For Southeast Asia, ozone enhancements are mostly driven by emissions within Southeast Asia, but we also see increasing contribution from South Asia (Fig.10c).

Climate change from 1995 to 2019 contributes substantially to the increase in tropospheric ozone over ESEA, as indicated by both GEOS-Chem and CMAQ model (Fig.7e). GEOS-Chem estimates that climate change has elevated the tropospheric ozone burden over ESEA by 1.5 Tg, accounting for 47% of the difference between 2019 and 1995. However, the magnitude of climate-driven TCO enhancement exhibits considerable variability in terms of spatial distribution and temporal evolution. Spatially, both models show that the largest climate-driven TCO increases are over the Qinghai-Tibet Plateau, central and southern China, and in the Indonesia. This spatial distribution is largely consistent with the spatial distribution of climate-driven surface ozone changes (Fig.11e), indicating that ground-level processes triggered by shifts in surface meteorological conditions (such as rise in surface temperature) play a crucial role in the climate-driven TCO changes. These surface processes will be described in the next section. Nevertheless, the TCO increase over Qinghai-Tibet Plateau implies that stratosphere-troposphere exchange (STE) of ozone, the key natural source of ozone in this region (Lu et al., 2019b; Chen et al., 2024), may have increased during 1995-2019, which contributes to increase tropospheric ozone over ESEA. We also find that the lightning NO emissions, the crucial natural ozone sources in the free troposphere, have been escalating by 17% over ESEA from 1995 to 2019 (Fig.S5). Lightning $NO_x$ emissions may also explain the higher estimation of climate-driven TCO increase in GEOS-Chem compared to the CMAQ model, as the CMAQ model does not account for lightning emissions. Stauffer et al. (2024) proposed that decreases in convective intensity and frequency facilitated ozone build-up in the free troposphere over equatorial Southeast Asia in the boreal spring. The ozone accumulation is also possible to propagate into summer and contributes to ozone increase.

## 5.1.2 Ground-level ozone

We now investigate the quantitative contribution of emission changes to summertime surface ozone trends from 1995 to 2019. As illustrated in Figure 11a, both the GEOS-Chem and WRF-CMAQ model simulate substantial surface ozone increases over continental ESEA in 2019 compared to the 1995 level, with an averaged enhancement of 7.8 ppbv in GEOS-Chem and 6.8 ppbv in WRF-CMAQ. Both models further confirm the dominant role of anthropogenic emissions in the surface ozone



enhancement (Fig.11b). The GEOS-Chem and WRF-CMAQ models simulate an ozone enhancement attributed to anthropogenic emission changes of 5.4 and 5.1 ppbv in 2019 compared to the 1995 level, averaged over continental ESEA, accounting for 69% and 75% of simulated ozone difference, respectively. This result indicates that the GEOS-Chem and WRF-CMAQ models exhibit higher consistency in attributing surface ozone trends in the ESEA region compared to their attribution of tropospheric ozone trends. Spatially, the emission-driven surface ozone enhancement can reach 10-20 ppbv in eastern and
central China, the Malay Peninsula, and the Korean Peninsula. These emission-driven ozone enhancements are also more pronounced at the surface compared to emission-driven TCO enhancement (Fig. 7).

Emission change within the ESEA appears to be the dominant driving factor of the rise in surface ozone levels across the region. As illustrates in Figure 11c, emission change within ESEA leads to substantial and continuous ozone enhancements in
most regions in China. The only exception is the NCP region, where both models indicate weak surface ozone increase in 2005 and 2013 compared to the 1995 level, and then ozone increase accelerates after 2013. This contrasts with the analysis for TCO, which shows that anthropogenic emissions consistently contribute to TCO increases over NCP from 1995 to 2013, while the increase slows down thereafter (Fig. 7).

The modest increase or even decline in summertime surface ozone concentrations, despite a significant rise in emissions in the NCP from 1995 to 2013, can be attributed to the $NO_x$-saturated regime for ozone production in this region. We examine the simulated changes in summertime surface ozone production efficiency (OPE), defined as the number of ozone molecules produced per molecule of $NO_x$ emitted, and the ratio of surface $H_2O_2$ to $HNO_3$ concentrations ($H_2O_2/HNO_3$) as indicators of the ozone chemical formation regime during 1995-2019 (Sillman, 1995; Wang et al., 2021). Figure 12 reveals that the NCP
region has the lowest $H_2O_2/HNO_3$ ratio and OPE values in ESEA due to the substantial anthropogenic and agricultural soil emissions, with values reaching their nadir in 2010 as anthropogenic $NO_x$ emissions began to decrease afterward (Fig.S8). As ozone chemical production is significantly restrained in such a $NO_x$-rich environment, the rapid and sustained increases in anthropogenic $NO_x$ emissions in the NCP region from 1995 to 2013 result in a much smaller ozone increase or even an ozone decrease compared to other regions such as the YRD. After 2013, the decrease in anthropogenic $NO_x$ emissions, coupled with
a slight rise in anthropogenic NMVOCs emissions, tends to elevate surface ozone levels, as will be discussed in Section 5.2. However, at higher altitudes over the NCP, ozone chemical production is more sensitive to $NO_x$ compared to that at the surface, hence ozone trends align with trends in anthropogenic $NO_x$ emissions. Our analysis clearly illustrates that emission changes in the NCP lead to a contrasting change in surface and tropospheric ozone, modulated by the chemical regime. This is also consistent with a recent study by Han et al. (2024), which shows contrasting response of surface and tropospheric ozone over
China to the emission reductions in 2013-2020.

There are also significant and sustained surface ozone increases driven by rising regional emissions in the Malay Archipelago, as simulated by both the GEOS-Chem and WRF-CMAQ models. In the Korean Peninsula and Japan, changes in anthropogenic





emissions lead to an increase in mean surface ozone by approximately 3-4 ppbv and 1-2 ppbv, respectively, from 1995 to 2019,
according to estimates from both models. These contributions are likely underestimated because the models tend to
underestimate long-term trends in surface ozone in both Japan and South Korea (Fig.6). However, this enhancement slows
down after 2013, coinciding with the observed ozone trends (Fig. 6), which may reflect the combined effect of emission
changes at both the domestic level and in the upwind NCP region.

Anthropogenic emissions originating from outside the ESEA also play a role in the surface ozone increase, resulting in an
averaged ozone enhancement of 0.8-1.1 ppbv across continental ESEA, 1-3 ppbv in western China, and 3-6 ppbv in the Malay
Archipelago (Fig.11d). A significant contributing source region appears to be India. Our analysis reveals that changes in
anthropogenic emissions outside ESEA lead to a summertime ozone increase of 6 ppbv in 2019 compared to the 1995 level
over India, and these increases are then transported to western China. However, the impact of anthropogenic emissions from
outside the ESEA on surface ozone is less pronounced than on tropospheric ozone (Fig.7) over eastern China, the Korean
Peninsula, and Japan, highlighting the shorter chemical lifetime of ozone in the polluted boundary layer compared to that in
the free troposphere. This is also supported by the analysis from the CAM4-chem tagged simulations (Fig.10).

Climate change explains about 25-30% of the simulated surface ozone difference averaged over the continental ESEA between
1995 and 2019, as indicated by both the GEOS-Chem and WRF-CMAQ models (Fig.11e). It increases surface ozone over
central and southern China, the Korean Peninsula and Indonesia by 6-10 ppbv in GEOS-Chem and 3-6 ppbv in WRF-CMAQ
from 1995 to 2019 effectively worsening surface ozone air quality in these regions. Analysis in Section 3.1 shows that
temperature has risen over the continental ESEA. The rise in temperature is anticipated to increase surface ozone
concentrations by boosting biogenic VOC emissions (Fig.S5), accelerating ozone chemical formation, and hindering dry
deposition. Decrease in wind speed is also favorable for ozone accumulation and contributes to increase in surface ozone
concentrations. The influence of climate change on surface ozone also shows significant interannual variability.

**5.2 2013-2019 ozone variability**

Section 5.1 discusses the factors contributing to tropospheric and surface ozone trends over ESEA in the long-term period of
1995-2019 based on chemical transport models. The same modeling framework can also be applied to investigate the
attribution of short-term interannual variability. However, since the interannual variability (IAV) tends to be more pronounced
than long-term trends, it is typically a challenge for chemical models to capture the observed magnitude of IAV. In this section,
we introduce statistical and machine learning models to attribute short-term ozone variability to anthropogenic emissions and
meteorological changes, and compare the results estimated from the GEOS-Chem and WRF-CMAQ chemical transport models.
We focus on our analysis in major Chinese city clusters, where summertime surface ozone shows extremely rapid increases in
2013-2019, despite the significant reduction in nationwide anthropogenic emissions of primary pollutants following the
implementation of the Chinese Action Plan on Air Pollution Prevention and Control since 2013. We focus on the MDA8 ozone



in the following discussion as it is one of the ozone air quality standard metrics in China and also an important metric of human health exposure.

### 5.2.1 Results from statistical and machine learning models

We start by examining the extent of ozone variability that can be captured using meteorological variables as predictors in three statistical and machine learning models (Section 2.2) at 74 cities in China with continuous monitoring sites since 2013. Figure 13 summarizes the coefficient of determination (R2) from all these algorithms using two sets of meteorological data. Overall, we find that the RFR model outperforms RR and MLR as evidenced by its relatively higher $R^2$ (0.46±0.16, mean±standard deviation across 74 cities, using ERA5 meteorological fields as predictors). This reflects the stronger capability of the RFR

model to adaptively capture both linear and nonlinear relationships between meteorological variables and ozone, compared to the other two models. We also find that, across all models, using ERA5 meteorological data as input yields higher predictive capability compared to MERRA2 when averaged across all cities. This may partly be attributed to the higher spatial resolution of ERA5 (0.25°×0.25°) compared to MERRA2 (0.5°×0.625°) for resolving the relationship between meteorological variables and ozone concentrations. However, $R^2$ is higher with MERRA2 than with ERA5 in southwestern China, including the SCB

city clusters.

We also find noticeable spatial variability in $R^2$ among cities. From all models using ERA5 data as input, $R^2$ is higher in the NCP (0.41-0.43), YRD (0.43-0.55), and PRD (0.54-0.64), while it is much smaller in the SCB (Table S3). This suggests that the observed ozone variability in eastern and southern China is, relatively, more strongly influenced by meteorological

variables compared to the western regions, where high background ozone levels may be less directly driven by local meteorology conditions (Lu et al., 2019b; Ye et al., 2024). Furthermore, complex terrain that is commonly found in these western regions, such as the SCB, may pose a greater challenge for models to achieve predictive skill equivalent to that in eastern and southern China.

Key meteorological variables influencing ozone variabilities of China are identified as temperature at 2m above the ground, relative humidity, solar radiation, and planetary boundary height (Fig.S9), with the most dominant variable expectedly varying across cities, in line with previous studies (Li et al., 2019b; Weng et al., 2022). Additional discrepancies in the identified key variables between models using ERA5 and MERRA2 are observed. For example, the predictor importance of solar radiation is substantially reduced when using MERRA2 (Fig.S9). The overall $R^2$ and importance of predictors are generally consistent

with Weng et al. (2022), which employs a similar modelling framework to fit surface ozone concentrations in China but for warm season (April-October) of 2015-2019 and only uses ERA5 data as input. Our analysis here focuses exclusively on the summertime period (June, July, August) while extending the study years to 2013–2019 and using both ERA5 and MERRA2 datasets (in separate regressions).



Figure 14 shows the 2013–2019 time series of both observed and modelled monthly MDA8 ozone anomalies in four city clusters, calculated as the differences between monthly values and their corresponding 2013–2019 monthly averages. The linear trends derived from the observations are 2.9, 1.7, 1.2, 1.4 ppbv year[-1] averaged over all cities in the NCP, YRD, PRD, and SCB regions, respectively. We also see large month-to-month variability superimposed on the trends. Much of the month-to-month variability can be captured by the statistical and machine learning model predictions, relying only on the

meteorological variables. Here, we assume a framework where the linear trends of the residual term, calculated as the difference between observed and predicted values, can serve as an approximation of ozone trends driven by changes in anthropogenic emissions.

We find that despite differences in predictive skills and the resulting key meteorological variables, all three models consistently

point to the dominant role of anthropogenic emissions in driving the 2013-2019 summertime surface ozone trend in the NCP, YRD, and PRD region. Our six estimates— three models with predictions from two meteorological inputs in turn—show that, changes in anthropogenic emissions cause ozone increases in the NCP at 2.4±0.1 ppbv year[-1] (mean±standard deviation across six estimates), accounting for 83% of the observed ozone trend. Changes in meteorology conditions contributes to the remaining 0.5±0.1 ppbv year[-1] (17%). Similar results are found in YRD, where the analysis attributes 81% (1.4±0.1 ppbv year[-1])

of the ozone increases to anthropogenic emission changes. The use of ERA5 or MERRA2 data only leads to slight differences (0.1 ppbv year[-1]) in trend attribution. Compared to the results of Li et al. (2020), our study attributes a greater portion of the ozone increase to changes in anthropogenic emissions in the NCP and YRD regions (81%-83% in our study vs 56-58% in Li et al. (2020)) for the same period. The primary reason for the disparity between the two studies lies in the difference in temporal resolution of the data used for modeling. Li et al. (2020) conducted MLR fitting based on a monthly anomaly data, whereas

our study operates on a daily scale, which utilizes much more records to fit ozone concentrations. In the PRD region, our results attribute 66% and 34% of the ozone increases to changes in anthropogenic emissions and meteorology, respectively.

For the SCB region, all estimates concur that anthropogenic emissions are the predominant contributors to the ozone increases, but the magnitude is largely different between ERA5 and MERRA2 datasets. Our method estimates that anthropogenic

emissions account for an ozone increase of 1.1 ppbv year[-1], representing 79% of the observed trend when using ERA5, but attribute all ozone increases to anthropogenic emissions when using MERRA2 as input. These results highlight the challenges and uncertainties associated with using meteorological fields to forecast ozone fluctuations in the SCB region with complex terrain. This difficulty is further evidenced by the relatively low predictive skills in the region (Figure 13).

**5.2.2 Results from chemical transport model and comparison with statistical/machine learning approaches**

Next, we examine the attribution of surface MDA8 ozone changes from 2013 to 2019 from GEOS-Chem and WRF-CMAQ chemical transport models, and compare the results with those obtained from statistical and machine learning models. Given that linear summertime trends derived from four-year simulations (2013, 2015, 2017, 2019) may not be robust, the results are



presented as the difference in ozone concentrations between two temporal segments, that is, the average of simulated values in summer of 2017 and 2019 compared to that of 2013 and 2015.


Figure 15 illustrates the spatial distributions of simulated ozone differences between the two periods (2017-2019 minus 2013-2015), and those contributed by meteorology and anthropogenic emissions, using the simulated ozone difference between the BASE and Fix_Globe_AC simulations. Both the GEOS-Chem and WRF-CMAQ models indicate an ozone increase in 2017-2019 compared to 2013-2015, yet both models underestimate the magnitude of this increase (Table 4). Specifically for the city

clusters, observations reveal an ozone enhancement of 14.3 ppbv, 8.6 ppbv, 4.5 ppbv, and 4.5 ppbv in the NCP, YRD, PRD, and SCB city clusters, respectively. In contrast, GEOS-Chem simulation only shows ozone increases of 3.1 ppbv, 6.0 ppbv, 1.7 ppbv, and 1.9 ppbv for the four city clusters, capturing only about 22%-70% of the observed enhancement. The WRF-CMAQ model demonstrates a better ability by capturing 46%, 81% of the observed ozone enhancement in the NCP and YRD, and fits well with the observed trends in SCB, but also poorly captures only 24% of the observed ozone enhancement in PRD.

This discrepancy underscores a significant challenge for chemical transport models in accurately capturing short-term ozone interannual variability, partly due to uncertainties in emission inventories and an underestimation of the model's sensitivity to meteorological parameters, as will be discussed later. In particular, we find that WRF-CMAQ model better captures (compared to GEOS-Chem) the ozone increases in SCB (104% vs 42%), possibly due to its finer resolution to resolve fine-scale meteorology in complex terrain. Finally, it is well-known that differences in the chemical mechanisms embedded in these two

models will likely also contribute to the discrepancies found here (e.g., Archibald et al., 2023; Li et al., 2019a; Weng et al.,2023).

Both the GEOS-Chem and WRF-CMAQ models indicate that changes in meteorological fields from 2013-2015 to 2017-2019 alone have resulted in a notable ozone enhancement of 2-6 ppbv in eastern and northern China (Fig.15b, e). These increases

are linked to a rise in surface temperature and solar radiation, coupled with a decrease in cloud cover, which are conducive to ozone chemical formation and accumulation (Fig. S10). Both models also demonstrate that shifts in meteorological conditions lead to a decrease in ozone levels over Southeast Asia. In terms of emissions, both the GEOS-Chem and WRF-CMAQ models concur on the ozone enhancement attributed to changes in anthropogenic emissions in eastern China, including the NCP and YRD regions, as well as in Southeast Asia (Fig.15c, f).


Figure 16 and Table 4 provide a summary of surface MDA8 ozone changes and their attribution across the city clusters from the two chemical models, and the comparison with the results from statistical and machine learning models. In the NCP region, GEOS-Chem and WRF-CMAQ model attribute 52% and 34% of the simulated ozone increase to changes in meteorological conditions, respectively. In comparison, the statistical and machine learning models estimate a mean contribution of only 14%

averaged over six estimates (three models run with two input meteorological data in turn). Correspondingly, contributions from changes in anthropogenic emissions are estimated as 48%, 66%, 86% from GEOS-Chem, WRF-CMAQ, and the mean of





statistical and machine learning model results, respectively. In the YRD region, contributions from changes in meteorological conditions to ozone enhancement are 64%, 44% and 25% from the three estimates, with the remaining 36%, 56%, and 75% contributed by changes in anthropogenic emissions. Thus, we find that all models consistently agree that the meteorological conditions from 2013 to 2019 have become more conducive to ozone formation in the NCP and YRD city clusters, thus enhancing the already positive trend. The statistical and machine learning models and WRF-CMAQ tend to attribute more ozone enhancement to changes in anthropogenic emissions compared to meteorological drivers, while GEOS-Chem results in a slightly higher attribution to meteorology, proportional to the ozone trends realized in the model in the first place. We will discuss the potential implications to the results later in this session.

We conclude that averaged over the three methods, anthropogenic emissions contribute to 66% and 56% of the summertime surface ozone enhancement during 2013-2019 in the NCP and YRD region, respectively, indicating that anthropogenic emissions are the primary driver of the recent ozone increase in these two city clusters.

In the PRD and SCB regions, we find that the GEOS-Chem, WRF-CMAQ, and the mean of statistical and machine learning models agree that changes in anthropogenic emissions make a larger contribution to ozone enhancement between 2017-2019 and 2013-2015, compared to contributions from meteorology. Further, the three machine learning models suggest a substantially smaller contribution of meteorology to the overall positive ozone trends in PRD and SCB, attributing only about 5% to the respective differences, with the remaining 95% attributable to changes in anthropogenic emissions. This is evident by changes in meteorological patterns. Figure S10 shows that there is much less notable enhancement in temperature and solar radiation in the PRD and SCB region between the 2017-2019 and 2013-2015 period, compared to the NCP and YRD region, suggesting that meteorological conditions have contribution impacts on ozone increase

We raise a number of factors that must be taken into account when interpreting the disparities in ozone change attribution between statistical/machine learning and chemical models. Similar to other studies using chemical transport models for separating the impact of emission and meteorology on ozone concentration, our study attributes ozone change driven by meteorological variables by fixing the anthropogenic level at the same year (1995 in this case) while using year-specific meteorological fields to drive the model. While this is a widely-used approach, it assumes that the chemical model can accurately simulate ozone sensitivity to changes in meteorological variables. However, the validity of this assumption is questionable. For examples, Yin et al. (2021) compares the correlation coefficients (r) between key meteorological variables and ozone concentration from observations and GEOS-Chem model prediction in China. They find that although the model can successfully capture the sign of correlation coefficients between ozone and these variables, there are large discrepancies in terms of the absolute values. Studies have shown that chemical transport models may underestimate the surface ozone-temperature sensitivity (e.g., Li et al., 2024; Wu et al., 2024). This indicates indicating that the model may lead to very incorrect estimates concerning the sensitivity of ozone to meteorological factors. An alternative way to attribute ozone concentration to





emissions and meteorological changes is to use a machine learning modeling approach to correct the observation-to-simulation bias before attribution of ozone change (Keller et al., 2021; Yin et al., 2021). In addition, the interannual variability in emission inventory may have large uncertainty that also influence the ozone attribution by chemical transport models. Such uncertainties might also be reflected in the fact that both GEOS-Chem and WRF-CMAQ substantially underestimate the observed rapid
enhancement of surface ozone concentration in China during 2013-2019.

In contrast to numerical models, statistical or machine learning methods aim to approximate the relationship between local meteorological factors and ozone. By design, a key challenge is to estimate the importance of meteorology through proxy variables that cannot be interpreted as true causal relationships, and which will integrate a number of direct (e.g., temperature-
dependence of ozone chemistry) and indirect (e.g., temperature-dependence of precursor emissions) effects. Similarly, they cannot resolve processes that are averaged out over the course of a day when using daily meteorological variables that are derived from the daytime window (06:00 to 18:00). Additionally, given that only local meteorological variables are considered here, the impact from larger-scale circulation or weather patterns may not be effectively captured by these methods. Nonetheless, despite these uncertainties, such approaches have the advantage of learning directly from observations, whereas
numerical models may be difficult to achieve precise reconstruction due to their inherent uncertainties. Furthermore, statistical or machine learning methods are more computation-friendly, particularly they do not require running perturbation simulations like numerical models here, which may also be subject to uncertainties.

Overall, it should be emphasized that there is no "best" model for trend attribution, as intrinsic uncertainties exist in both
numerical models and the data-driven statistical or machine learning methods. Nonetheless, by using a wide range of models with significantly different characteristics, our study provides multiple lines as evidence, which, in our view, is a good practice to address the uncertainty associated with any single method or model.

### 5.2.3 Review of the mechanisms contributing to summertime surface ozone increase in China from 2013 to 2019

Previous studies elucidated the mechanisms through which changes in anthropogenic emissions across China have contributed to the enhancement of ozone levels in 2013-2019 (Wang et al., 2022c). Since 2013, the Chinese government has implemented the Air Pollution Prevention and Control Action Plan (2013−2017) and the Three-Year Action Plan for Winning the Blue Sky Defense Battle (2018−2020). These initiatives have led to prominent reductions in anthropogenic emissions of $NO_x$ (21% between 2013-2017 and 25% between 2013-2019) and CO. In contrast, NMVOCs emissions showed a slight increase (7%)
from 2013-2015, followed by a minor decline (4%) from 2015, resulting in an overall flat trend between 2013-2019 (-1%). The simultaneous decrease in $NO_x$ and increase in NMVOCs emissions are expected to elevate ozone concentrations in regions such as the NCP, where ozone chemical productions are prone to being VOC-limited or mixed-sensitive (Wang et al., 2023a;



Wang et al., 2023b). In the PRD region where $NO_x$ emissions began to decline earlier than NCP and YRD, study pointed out that long-term ozone trends was dominantly driven by the decreased titration effect by NO (Li et al., 2022).


Furthermore, the rapid decline in particulate matters (PM), facilitated by policy-driven reductions in precursor emissions, can increase ozone due to a reduced loss of hydroperoxy radicals ($HO_2$) and increased photolysis (Gao et al., 2020b). A number of studies show that the impact of reduced heterogeneous uptake of $HO_2$ radicals on the ozone trend is more significant than the effects of increased photolysis rate and reduced $NO_x$ concentration between 2013-2017, although the dominant process may

vary regionally (Li et al., 2019b,c; Liu and Wang et al., 2020b; Shao et al., 2021; Liu et al., 2023). However, several studies contend that there is insufficient observation-based evidence to support the importance of heterogeneous chemistry on radical concentrations (e.g., Tan et al., 2019). A recent study also pointed out that increase ozone production efficiency from agriculture soil emissions with the reduced anthropogenic $NO_x$ emissions may have also contributed to ozone increase in the NCP region (Tan et al., 2023). A key policy-relevant conclusion drawn from these studies is that, while nationwide control

measures from 2013 to 2017 have successfully alleviate $PM_{2.5}$ pollution, they have also led to increased $O_3$ concentrations in urban areas of China, due to the non-linear dependence of $O_3$ on $NO_x$ and aerosol feedbacks. New analysis for ozone trends after 2018 suggests that synergistic control of VOC and $NO_x$ has started to be effective to mitigate ozone pollution, as $PM_{2.5}$ has become much lower compared to 2013-2017 level (Yin et al., 2021; Liu et al., 2023; Wang et al. 2023b; Wang et al., 2024).

The mechanisms by which changes in atmospheric circulation or local meteorological elements since 2013 exacerbated summertime surface ozone pollution in China have been extensively studied (e.g., Lu et al., 2019b; Liu and Wang, 2020; Dang et al., 2021; Gong et al., 2022b; Weng et al., 2022; Kou et al., 2023). Summertime meteorological conditions trend towards rising temperatures and increased solar radiation since 2013 (Fig.S10). These changes in meteorological variables are highly conducive to ozone formation by enhancing photochemical production and suppressing ozone ventilation. The dominant

meteorological drivers, as well as the weather system and synoptic patterns they reflect, vary in different regions of China. For instance, Li et al. (2020) shows that rising summertime temperatures associated with increased foehn winds are the main drivers of $O_3$ increase over the NCP. On a regional scale, the favorable meteorological conditions are controlled and modulated by specific weather systems, such as the Western Pacific Subtropical High, periphery of tropical cyclones, and anticyclone (Gong et al., 2022a; Shu et al., 2020; Wang et al., 2022c; Hu et al., 2024). For examples, in the PRD region, Liu et al. (2025)

show that the increased frequency of ozone-favorable weather patterns, including the periphery of tropical cyclones and the influence of the WPSH, has significantly contributed to the observed ozone trends. These findings underscore the importance of both large-scale atmospheric circulation changes and local meteorological factors in driving short-term interannual variability in ozone pollution.





## 6 Discussion and Conclusions

Observations have revealed substantial ozone increases both in the free troposphere and at the surface over most regions in East and Southeast Asia from 1995. In this study, we apply a statistical model (multiple linear regression), two machine learning models (ridge regression and random forest regression), and three chemical transport models (GEOS-Chem, CAM4-chem, and WRF-CMAQ) to attribute long-term (1995-2019) and/or short-term (2013-2019) ozone trends to changes in anthropogenic emissions and climate, spanning from surface to tropopause across ESEA. Anthropogenic emissions of $NO_x$,

CO, and NMVOCs have increased by 129%, 17%, and 50% from 1995 to 2019 over the continental ESEA, respectively. Summertime surface temperature has increased by 1.5%, which enhances biogenic VOCs emissions by 13%. The comparison with observations indicates that the GEOS-Chem model fits well with the tropospheric ozone profiles, but severely overestimates surface ozone. The WRF-CMAQ model performs well in simulating surface ozone concentration levels but has significant biases in tropospheric ozone simulation. The CAM4-chem model provides satisfactory results for both tropospheric

and surface ozone simulations, albeit with a coarser resolution. These three chemical transport models have reproduced the observed tropospheric ozone increases from 1995 but underestimate the magnitude of trends.

We find that all three chemical transport models, while with varied characteristics in the model capability and emission inventory, agree that changes in anthropogenic emission drive ozone increases both in the free troposphere and at the surface

from 1995 to 2019. In the free troposphere, GEOS-Chem and WRF-CMAQ models attribute 53% and 59% increase in tropospheric ozone burden to changes in anthropogenic emissions from 1995 to 2019. Emission changes inside ESEA contribute to 66-77% of the emission-driven TCO increases over ESEA. However, after 2013, TCO produced from ESEA anthropogenic emissions flattens or even decreases, largely due to emission reduction in China. In contrast, emissions outside ESEA continue contributing to TCO increases in ESEA, with ozone contributions being larger with higher altitudes. The

tagged simulation in CAM4-chem confirms that South Asia, in particular India, has an increasing contribution to TCO increases in ESEA, while Europe and North America show decreasing contribution due to the control of anthropogenic emissions in these regions. At the surface, the GEOS-Chem and WRF-CMAQ model attribute 69% and 75% ozone increase to changes in anthropogenic emissions, mostly from anthropogenic emission changes within ESEA. In particular, we find that emission reduction in China after 2013 has led to a different response of ozone at the surface (increase) and in the troposphere

(decrease). Climate change from 1995 to 2019 also contributes substantially to the increase in summertime tropospheric (41-47%) and surface ozone (25-31%) over ESEA. These climate-driven ozone increases may associate with surface warming that triggers more active natural emissions and photochemistry.

Multiple methods have been applied to attribute the surface ozone change from 2013 to 2019 in China, where observed ozone

shows rapid increases during the period in major city clusters (1.2-2.9 ppbv year$^{-1}$). In the NCP region, the mean of statistical and machine learning model results, GEOS-Chem, and WRF-CMAQ model estimate that 86%, 46%, 66% of the surface ozone

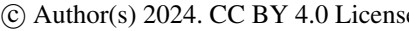



increase is attributed to anthropogenic emissions changes. While the overall direction is consistent, the range of the estimates reflect discernable uncertainty from each model. We conclude that averaged over the three methods, anthropogenic emissions contribute to 66% and 56% of the summertime surface ozone enhancement from 2013-2019 in the NCP and YRD region,

respectively, indicating that anthropogenic emissions are the key driver of the recent ozone increase in China. The rest of the ozone increases are contributed by meteorological patterns, supported by the enhanced surface temperature and solar radiation. In contrast, we find that changes in anthropogenic emissions dominate summertime surface ozone increase in the PRD and SCB (about 95%) regions, with minor contributions (5%) from changes in meteorological conditions.

Our study thus provides a quantitative attribution of tropospheric ozone trends ESEA over both a long-term (1995-2019) and short-term (2013-2019) periods. These results highlight significant ozone increases contributed by enhanced anthropogenic emissions, with additional contribution from climate change. There are notable limitations in this project that require much future work.

First, we note that chemical transport models struggle to reproduce long-term trends and short-term variability in ozone over ESEA, especially at the surface level. This remains a major limitation in this study, and leads to significant uncertainties in the attribution of ozone trends. From an observational perspective, we call for increased availability of stable and long-term observations over ESEA, where ozone concentrations exhibit intense spatial and temporal heterogeneity and variability, to provide sufficient information for validating the performance of chemical models, thereby facilitating the improvement of

these models. From a modeling perspective, there is a need for more precise and higher-resolution emission inventories of ozone precursors, particularly those that better constrain anthropogenic emissions in Southeast Asia before 2010. Additionally, increasing model resolution is crucial for simulating ozone distributions over urban clusters in ESEA.

Second, inherent limitations within each model contribute to substantial variations in the attribution of ozone trends. It is also

challenging to assess what causes the inconsistencies between models. This difficulty primarily reflects the coupled nonlinear impact of emissions, chemistry, and meteorology on ozone. Therefore, considering the combination of different modeling approaches, such as integrating process-based chemical models with data-driven deep learning methods, is a viable option for providing a more robust ozone attribution (Keller et al., 2021; Yin et al., 2021; Liu et al., 2022). Future efforts should also be placed in incorporating more numerical models with unified emission datasets to better specify the causes of in model

inconsistency of trend attribution.

Third, although our work has quantified the contributions of emissions and meteorology to rising ozone levels, it has not conducted an in-depth analysis of the mechanisms that have been proposed in existing studies. Future work needs to combine more observational data (such as concentrations of ozone precursors, particularly VOCs) and model analyses to disentangle

the mechanisms driving ozone increases in different regions, including the transboundary transport within ESEA.

Fourth, due to significant computational costs, this study focused only on summertime ozone trends. However, observational data indicate that tropospheric ozone increases in East Asia during the cold season have also been significant in recent years (Gaudel et al., 2020). At the surface level, several studies have reported increases in springtime surface ozone levels in China

(Li et al., 2021; Cao et al., 2024). In addition, our focus has been placed on surface MDA8 ozone concentrations, yet recent studies have also revealed increased nighttime atmospheric oxidation capacity and ozone concentration especially in China (He et al., 2022; Wang et al., 2023c; He et al., 2023). Further research and attention are needed in these contexts in the future.

**Data availability**

All data used in this study, including the observations and model results, will be archived in open-access data portal upon the

publication of the manuscript.

**Supplement**

The supplement related to this article is available online.

**Author Contribution**

Xiao Lu and Tatsuya Nagashima, co-leads of the East Asia Working Group of Tropospheric Ozone Assessment Report Phase

II (TOAR II), led and organized the project. Xiao Lu, Jiayin Su, Haolin Wang, Jingyu Li, Cheng He, and Shuai Li conducted the GEOS-Chem simulations. Yiming Liu and Yuqi Zhu conducted the WRF-CMAQ simulations. Tabish Ansari and Tim Butler conducted the CAM4-chem simulations. Yuqiang Zhang conducted another set of CAM-chem simulations for this study. However, the result was not analyzed due to its similarity with the existing CAM4-chem results. Xiang Weng and Ganquan Zeng conducted the statistical and machine learning models. Teerachai Amnuaylojaroen, Mohd Talib Latif, Xiaobin Xu, Ja-

Ho Koo, and Tatsuya Nagashima assisted in preparation of surface observational data. Philippe Nédélec and Bastien Sauvage contributed the IAGOS data and advised on its interpretation. Xiao Lu, Yiming Liu, Jiayin Su, Xiang Weng, Tabish Ansari, Guowen He, Haolin Wang, and Ganquan Zeng analyzed the results and prepared the figures. Xiao Lu wrote the manuscript with significant contributions from Yiming Liu, Xiang Weng, and Tabish Ansari. All authors contributed to the interpretation of the results and revision of the manuscript.

**Competing interests**

Some authors are members of the editorial board of Atmospheric Chemistry and Physics.



**Acknowledgement**

MOZAIC/CARIBIC/IAGOS data were created with support from the European Commission, national agencies in Germany (BMBF), France (MESR), and the UK (NERC), and the IAGOS member institutions (http://www.iagos.org/partners). The

participating airlines (Lufthansa, Air France, Austrian, China Airlines, Hawaiian Airlines, Air Canada, Iberia, Eurowings Discover, Cathay Pacific, Air Namibia, Sabena) supported IAGOS by carrying the measurement equipment free of charge since 1994. The data are available at http://www.iagos.fr thanks to additional support from AERIS.

**Financial support**

This research has been supported by the National Key Research and Development Program of China (2023YFC3706104), the

National Natural Science Foundation of China (grant no. 42105103), and the Young Elite Scientists Sponsorship Program by CAST (2023QNRC001).

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

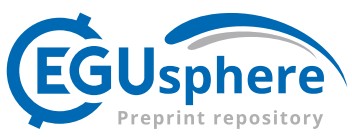

**Table 1: Summary of ozone observations used in this study.**

### National surface monitoring network

| Region | Network | Number of sites used in this study | Period | Data source (website) |
|---|---|---|---|---|
| China | China National Environmental Monitoring Centre | 1361 sites | 2013-2019 | https://106.37.208.233:20035/ |
| | | 72 sites in BTH, 159 sites in YRD, 54 sites in PRD, and 84 in SCB | | |
| Hong Kong (China) | Environmental Protection Department, the Government of the Hong Kong Special Administrative Region, China | 9 sites | 2000-2019 | https://cd.epic.epd.gov.hk/EPICDI/air/station/?lang=zh |
| Republic of Korea | Korean Ministry of Environment | 289 | 2001-2019 | https://airkorea.or.kr/web/ |
| Japan | National Institute for Environmental Studies | 800 | 1995-2019 | https://tenbou.nies.go.jp/download/ |
| Malaysia | Continuous Air Quality Monitoring Stations | 13 | 2017 | From Mohd Talib Latif |
| Thailand | Thailand Pollution Control Department | 11 | 2005-2019 | https://www.pcd.go.th/ |

### Individual surface monitoring sites

| Site name | Region | Latitude | Longitude | Elevation/m | Period | Data source (website) |
|---|---|---|---|---|---|---|
| Longfenshan | China | 44.37°N | 127.6°E | 330.5 | 2005-2016 | |
| LinAn | | 30.3°N | 119.44°E | 139 | 2005-2016 | |
| Shangdianzi | | 40.65°N | 117.17°E | 293.9 | 2003-2016 | |
| China Meteorological | | 39.95°N | 116.32°E | 96 | 2008-2018 | https://join.fz-juelich.de/access/db/ |
| GuCheng | | 39.13°N | 115.12°E | 15.2 | 2006-2018 | |

### Ozonesonde observations

| Site | Region | Latitude | Longitude | Elevation/m | Period | Number of available profiles | Data source (website) |
|---|---|---|---|---|---|---|---|
| Pohang | Republic of Korea | 36.0320°N | 129.3800°E | 3.94 | 1995-2019 | 245 | https://woudc.org/data.php |





| Station | Country | Latitude | Longitude | | Period | Number of available profiles | Data source (website) |
|---|---|---|---|---|---|---|---|
| Naha | Japan | 26.2072°N | 127.6875°E | 28.06 | 1995–2017 | 231 | |
| Tateno (Tsukuba) | Japan | 36.0581°N | 140.1258°E | 25.2 | 1995–2019 | 262 | |
| Sapporo | Japan | 43.0600°N | 141.3286°E | 17.45 | 1995–2019 | 245 | |
| King's Park | Hong Kong, China | 22.3118°N, | 114.1728°E, | 65 | 2000–2017 | 210 | |
| Hanoi | Viet Nam | 21.0215°N | 105.8731°E | 6 | 2005–2019 | 74 | |
| Kuala Lumpur | Malaysia | 2.7°N | 101.7°E | 10 | 1998–2019 | 110 | https://tropo.gsfc.nasa.gov/shadoz/Kuala.html |

IAGOS region

| Region | Range | Number of available profiles | Data source (website) |
|---|---|---|---|
| East Asia | 27°N–46°N, 103°E–137°E | 2584 | https://www.iagos.org/ |
| Southeast Asia | –10°S–27°N, 90°E–123°E | 2059 | |




**Table 2: Configuration of simulations used in this study.**

| Simulation Name | Description |
|---|---|
| BASE | Simulations with year-specific anthropogenic emissions, methane concentrations, and meteorological fields. Conducted by a GEOS-Chem and WRF-CMAQ model. |
| Fix_Globe_AC | Same as BASE, but with global anthropogenic emissions and methane concentration fixed at their 1995 levels. Conducted for GEOS-Chem and WRF-CMAQ model. |
| Fix_ESA_A | Same as BASE, but with anthropogenic emissions fixed at their 1995 levels. Conducted for GEOS-Chem and WRF-CMAQ model. |
| NO$_x$-tagged and RC-tagged | Simulations with year-specific anthropogenic emissions, methane concentrations, and meteorological fields, at the same time attribute the simulated ozone in terms of its NO$_x$ and reactive carbon (RC, NMVOC+CO+CH$_4$) sources. Conducted for CAM4-chem model. |

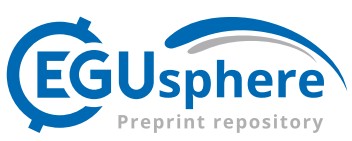

**Table 3: Summary of the characteristics of the three chemical transport model used in this study.**

| | GEOS-Chem | CAM4-chem | WRF-CMAQ |
|---|---|---|---|
| Meteorology input. | Using MERRA-2 reanalysis data as offline input. | Driven by meteorological data from the MERRA2 reanalysis, with some fields from MERRA2 reanalysis variables being nudged every time step (30 min) by 10 % towards analysis fields. | Using WRF simulated meteorological fields as offline input. |
| Chemistry | UCX scheme, with full tropospheric and stratospheric chemistry. Methane boundary conditions from observations with interannual variability | MOZART-4 tropospheric chemical mechanism, simplified stratospheric chemistry. Methane concentrations imposed as boundary conditions. | SAPRC07TIC for gas-phase chemistry and AERO6i for aerosol mechanism, no stratospheric chemistry. Methane concentration fixed as a constant. |
| Emissions | CEDSv2 anthropogenic emissions, including aircraft emissions. Natural emissions include biogenic VOCs, soil $NO_x$, lightning $NO_x$, BB4CMIP6 inventory for biomass burning emissions (1997-2017 same as GFED4); | CEDS$_{CMIP}$ anthropogenic emissions. Natural emissions include biogenic VOCs, soil $NO_x$, lightning $NO_x$, GFED4 inventory for biomass burning emissions. | MIX inventory with interannual variations being scaled by the CEDSv2 inventory. No aircraft emissions. Natural emissions include biogenic VOCs, soil $NO_x$, GFED for biomass burning emissions. No lightning $NO_x$ emissions. |
| Resolution and domain | Global: 4°×5°, with 72 vertical layers extending to 0.01 hPa East and Southeast Asia: 0.5°× 0.625°, with 72 vertical layers | 1.9°×2.5°, with 56 vertical levels | 36 km × 36 km, with 23 vertical layers extending to the height of ~22 km |
| Simulation time | 1995-2019 for BASE simulation, June, July, and August in 1995, 2000, 2005, 2010, 2013, 2015, 2017, 2019 for sensitivity simulations | 2000-2018 | June, July, and August in 1995, 2000, 2005, 2010, 2013, 2015, 2017, 2019 for sensitivity simulations |




**Table 4: Meteorological and anthropogenic emission driven summertime ozone difference between 2017-2019 and 2013-2015 over the key four city clusters in China.**

| | | MLR | RR | RFR | GEOS-Chem | WRF-CMAQ |
|---|---|---|---|---|---|---|
| **NCP** | | | | | | |
| **Obs/Base[a]** | | 14.3 | 14.3 | 14.3 | 3.1 | 6.6 |
| **Met[b]** | MERRA2[d] | 2.3 (16.1%) | 1.9 (13.3%) | 1.9 (13.3%) | 1.6 (51.6%) | 2.2 (33.9%) |
| | ERA5[e] | 2.1 (14.7%) | 1.7 (11.9%) | 2.3 (16.1%) | | |
| **Anth[c]** | MERRA2 | 12.0 (83.9%) | 12.4 (86.7%) | 12.4 (86.7%) | 1.4 (48.4%) | 4.3 (66.1%) |
| | ERA5 | 12.2 (85.3%) | 12.6 (88.1%) | 12.0 (83.9%) | | |
| **YRD** | | | | | | |
| **Obs/Base** | | 8.6 | 8.6 | 8.6 | 6.0 | 7.0 |
| **Met** | MERRA2 | 3.4 (39.5%) | 3.1 (36.0%) | 2.6 (30.2%) | 3.9 (64.4%) | 3.1 (44.0%) |
| | ERA5 | 1.0 (11.6%) | 0.9 (10.5%) | 1.8 (20.9%) | | |
| **Anth** | MERRA2 | 5.2 (60.5%) | 5.5 (64.0%) | 6.0 (69.8%) | 2.14 (35.6%) | 3.9 (56.0%) |
| | ERA5 | 7.6 (88.4%) | 7.7 (89.5%) | 6.8 (79.1%) | | |
| **PRD** | | | | | | |
| **Obs/base** | | 4.5 | 4.5 | 4.5 | 1.7 | 1.1 |
| **Met** | MERRA2 | 1.6 (35.6%) | 1.4 (31.1%) | 0.9 (20.0%) | 0.4 (23.8%) | -0.4 (-32.7%) |
| | ERA5 | 0.7 (15.6%) | 0.7 (15.6%) | 0.7 (15.6%) | | |
| **Anth** | MERRA2 | 2.9 (64.4%) | 3.1 (68.9%) | 3.6 (80.0%) | 1.3 (76.2%) | 1.4 (132.7%) |
| | ERA5 | 3.8 | 3.8 | 3.8 | | |



| | | (84.4%) | (84.4%) | (84.4%) | | |
|---|---|---|---|---|---|---|
| **SCB** | | | | | | |
| **Obs/base** | | 4.5 | 4.5 | 4.5 | 1.9 | 4.7 |
| **Met** | MERRA2 | -1.4 (-31.1%) | -1.3 (-28.9%) | -1.2 (-26.7%) | 0.5 (25.5%) | 0.5 (11.2%) |
| | ERA5 | -0.2 (-4.4%) | -0.3 (-6.7%) | -0.4 (-8.9%) | | |
| **Anth** | MERRA2 | 5.9 (131.1%) | 5.8 (128.9%) | 5.7 (126.7%) | 1.4 (74.5%) | 4.1 (88.8%) |
| | ERA5 | 4.7 (104.4%) | 4.8 (106.7%) | 4.9 (108.9%) | | |

[a] For MLR, RR, RFR models, values are observed ozone difference between the two periods. For GEOS-Chem and CMAQ model, values are simulated ozone difference between the two periods.

[b] Met represents the meteorology-driven ozone difference. See the text for the calculation method. The values in the parenthesis are the percentage to the observed or simulated total ozone difference.

[c] Anth represents the anthropogenic emission-driven ozone difference. See the text for the calculation method. The values in the parenthesis are the percentage to the observed or simulated total ozone difference.

[d] Results using MERRA2 meteorological fields as input.

[e] Results using ERA5 meteorological fields as input.

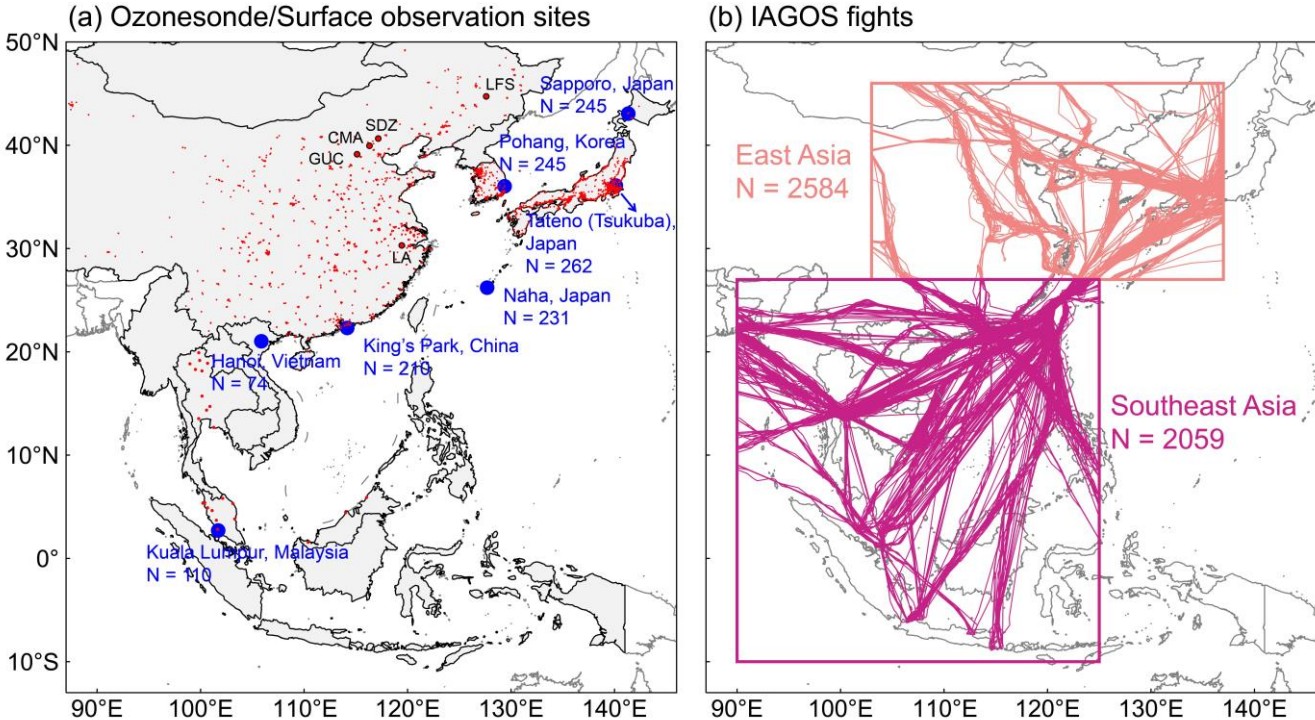

**Figure 1: Tropospheric and surface ozone observations from the IAGOS, ozonesonde, and surface monitoring networks used in this study in summertime 1995-2019. Panel (a) shows the location of surface monitoring sites (red) and ozonesonde sites (blue). The small red dots represent monitoring sites from the national network, the large red dots represent the five monitoring sites in China with long-term ozone observations. Numbers of available profiles (N) for ozonesonde measurements are shown inset. Panel (b) shows the IAGOS flight tracks in the troposphere. The boxes indicate the region of East Asia (103-137°E, 27-46°N) and Southeast Asia (90-123°E, -10-27°N). Numbers of available profiles (N) for flight tracks are shown inset. Detailed information of the observations is summarized in Table 1. Area with grey shadings in Panel (a) denote ESEA countries defined in this study (Mongolia, China, Democratic People's Republic of Korea, Republic of Korea, Japan, Myanmar, Thailand, Laos, Viet Nam, Philippines, Cambodia, Malaysia, Indonesia).**





**Figure 2: Spatial distributions and long-term trends in summertime anthropogenic emissions of NO$_x$, CO, and NMVOC in East and Southeast Asia. Emission estimates are from the CEDSv2 inventory. Panels (a1-a3) show mean emissions averaged over years 2015, 2017, and 2019. Panels (b1-b3) show the 1995-2019 trends. Black dots denoted linear trends with a p-value<0.05. Panels (c1-b3) show the time series of emission ratio relative to 1995 level for different regions, in which CHN stands for China, JP stands for Japan, S.K. stands for South Korea, SEA includes Myanmar, Thailand, Laos, Viet Nam, Cambodia, and Philippines, I/M stands for Indonesia and Malaysia.**







**Figure 3: Evaluation of GEOS-Chem, WRF-CMAQ, and CAM4-chem model simulated summertime tropospheric ozone distributions over IAGOS regions and at ozonesonde sites. Results are presented as averages for June, July, and August in 2015 and 2017 (representing present-day level), when output are available from all three models. Panels (a) and (b) are comparisons for the IAGOS regions defined in Figure 1b. The solid lines are observed and simulated ozone profiles along the IAGOS flight tracks. The dash lines (only shown for CAM4-chem and GEOS-Chem at 0.5°×0.625° resolution) are regional mean over the East Asia and Southeast Asia domain (Figure 1b), as CAM4-chem does not output hourly ozone for direct comparison with the IAGOS observations. Panels (c) and (d) are comparisons for the ozonesonde profiles. Horizontal bars represent standard deviation from observations at each vertical layers with an interval of 25 hPa. Numbers of available profiles (N) for comparison are shown inset.**







**Figure 4: Evaluation of GEOS-Chem, WRF-CMAQ, and CAM4-chem model simulated summertime surface MDA8 ozone concentrations. Results are presented as averages for June, July, and August in 2017 (representing present-day level). Panel (a) shows the distributions of observed ozone. Panels (b-d) are same as Panel (a), but for simulated ozone GEOS-Chem at fine (0.5°×0.625°) resolution, WRF-CMAQ model, and CAM4-chem model, respectively. Mean values ± standard deviation across different regions are shown inset. The spatial correlations (r) between the observation and simulation are also shown for GEOS-Chem (0.5°×0.625°) and WRF-CMAQ model, while for the CAM4-chem models r is not shown as the spatial resolution is too coarse to resolve the ozone deviation at different sites.**




Annual trends of the 50th percentiles of ozone, IAGOS vs GEOS-Chem (1995-2019)

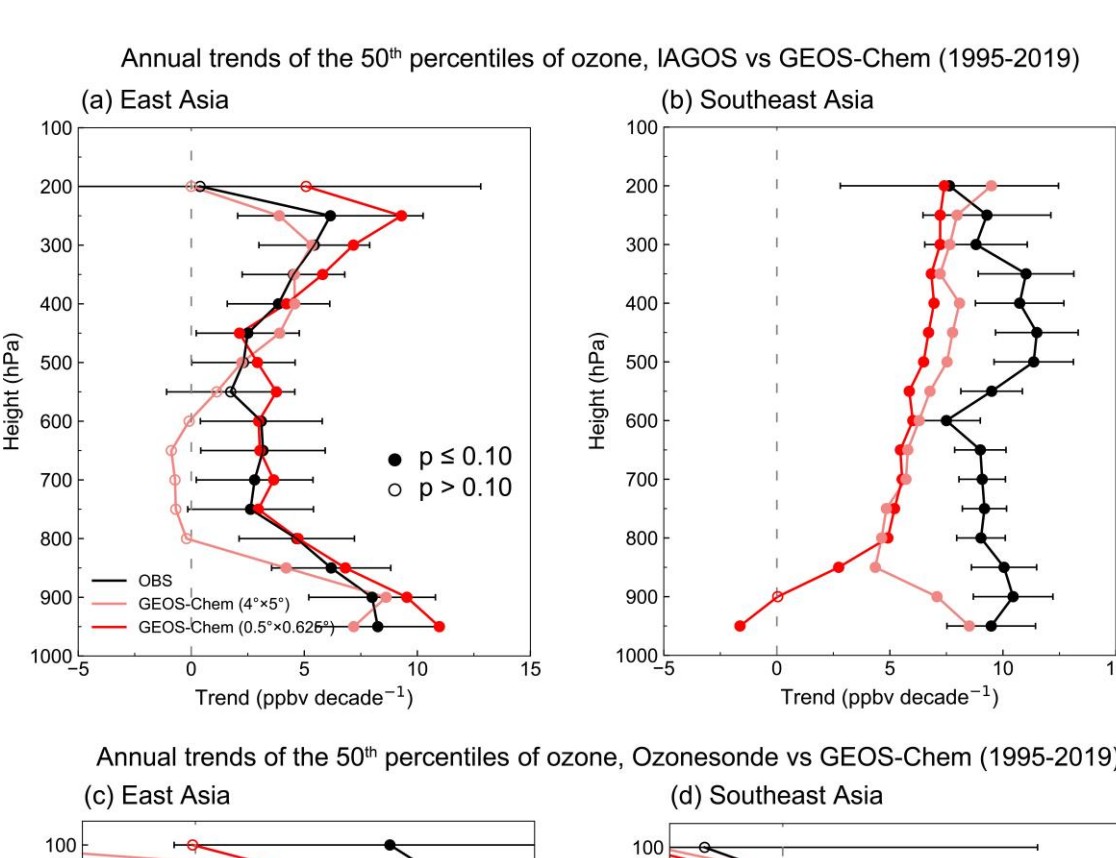

Annual trends of the 50th percentiles of ozone, Ozonesonde vs GEOS-Chem (1995-2019)

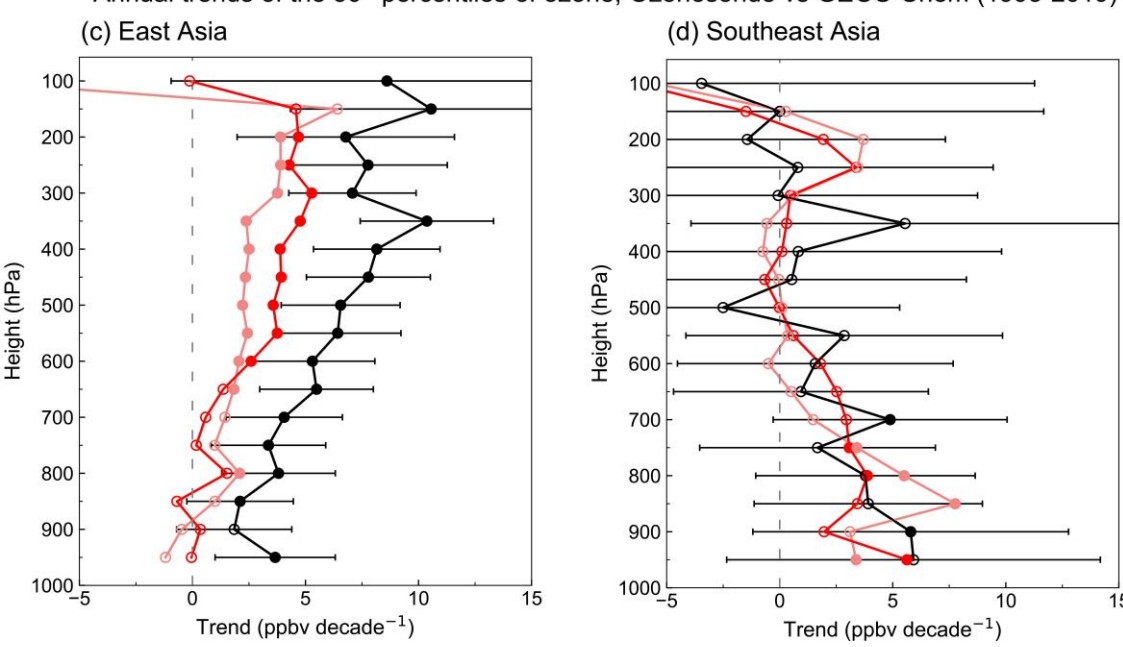

**Figure 5: Evaluation of GEOS-Chem's ability to capture summertime tropospheric ozone trends over IAGOS regions and at ozonesonde sites. Panel (a) shows trends of the 50th percentiles of IAGOS observed and GEOS-Chem simulated summertime ozone trends (ppbv per decade) at intervals of 50 hPa in 1995-2019. The trends are calculated using the quantile regression method (Section 2.4). Filled circles indicate trends with p-value<0.10. Horizontal bars represent trends at 90% confidence level from observations.**





**Figure 6: Evaluation of GEOS-Chem, WRF-CMAQ, and CAM4-chem model ability to capture 1995-2019 summertime surface ozone trends. The selected periods for each region/site correspond to the overlapping years of available observations and simulations. Ozone trends derived from the observations and different models, the associated p-values, and the correlation coefficients between the observed and simulated values are shown inset.**



**Figure 7: Factors contributing to changes in summertime tropospheric ozone column (represented as column-averaged in unit of ppbv) in East Asia and Southeast Asia estimated from the GEOS-Chem and WRF-CMAQ model. Results are ozone differences between the corresponding year and 1995. Ozone differences contributed by changes in global anthropogenic emissions (including surface emissions, aircraft emissions, and methane) (Panel b), anthropogenic emissions from ESEA (Panel c), anthropogenic emissions from outside ESEA (Panel d), and climate (including biomass burning and stratospheric influences) relative to 1995 (Panel e) are estimated. Numbers are mean value across the continental ESEA.**



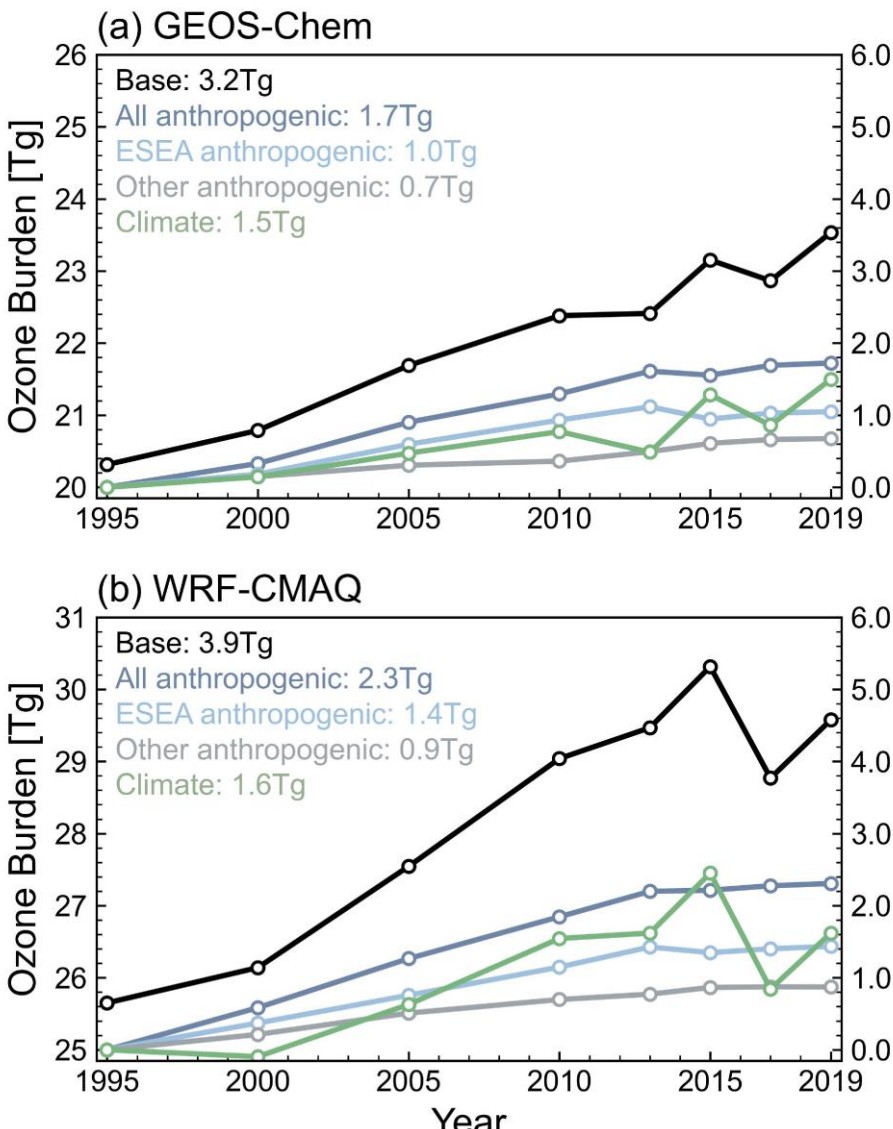

**Figure 8: Attribution of summertime tropospheric ozone (represented as tropospheric ozone burden in unit of Tg) in ESEA (including East Asia domain of 80°E-145°E, 30°N-53°N and Southeast Asia domain of 92.5°E-135°E, 10°S-30°N). Results are estimated from the GEOS-Chem and WRF-CMAQ model. Tropospheric ozone burden from the BASE simulation (left y-axis) are absolute values from 1995 to 2019. Changes in tropospheric burden attributed to global anthropogenic emissions (including surface emissions, aircraft emissions, and methane), anthropogenic emissions from ESEA, anthropogenic emissions from other regions, and climate (including biomass burning and stratospheric influences) are values relative to 1995 (right y-axis).**



**Figure 9: Vertical distribution of ozone difference contributed by anthropogenic emissions from ESEA (black) and from outside ESEA (grey) between 2019 and 1995 level, estimated from GEOS-Chem and WRF-CMAQ model. Panels (a), (b), and (c) show the results average over the ESEA domain (including East Asia domain of 80°E-145°E, 30°N-53°N and Southeast Asia domain of 92.5°E-135°E, 10°S-30°N), East Asia domain, and Southeast Asia domain, respectively.**



**Figure 10: Regional contribution to tropospheric ozone column and surface ozone in East Asia and Southeast Asia in 2000-2018. Results are estimated from a tagged module implemented in the CAM4-chem model (2.3.2). Definition of the regions are shown in Figure S3. Panels (a) and (b) are for tropospheric column ozone and surface ozone, respectively. Each line represents the ozone contribution to East Asia from ozone produced by anthropogenic nitrogen oxide emissions in a specific region. Panels (c) and (d) are the same as (a) and (b), but for Southeast Asia.**







Figure 11: Same as Figure 7 but for surface ozone.





**Figure 12: Changes in surface ozone chemical formation regime and ozone production efficiency and at surface in summer 1995-2019. Ozone chemical formation regime is examined using the ratio of H₂O₂ to HNO₃ concentration at the surface. Ozone production efficiency (OPE) is defined as the number of ozone molecules produced per molecule of NOₓ emitted, and is only diagnosed in GEOS-Chem. Panels (a1), (b1), and (c1) show the spatial distributions in summer 2013, and the rest panels show the difference relative to 2013 level.**

1490





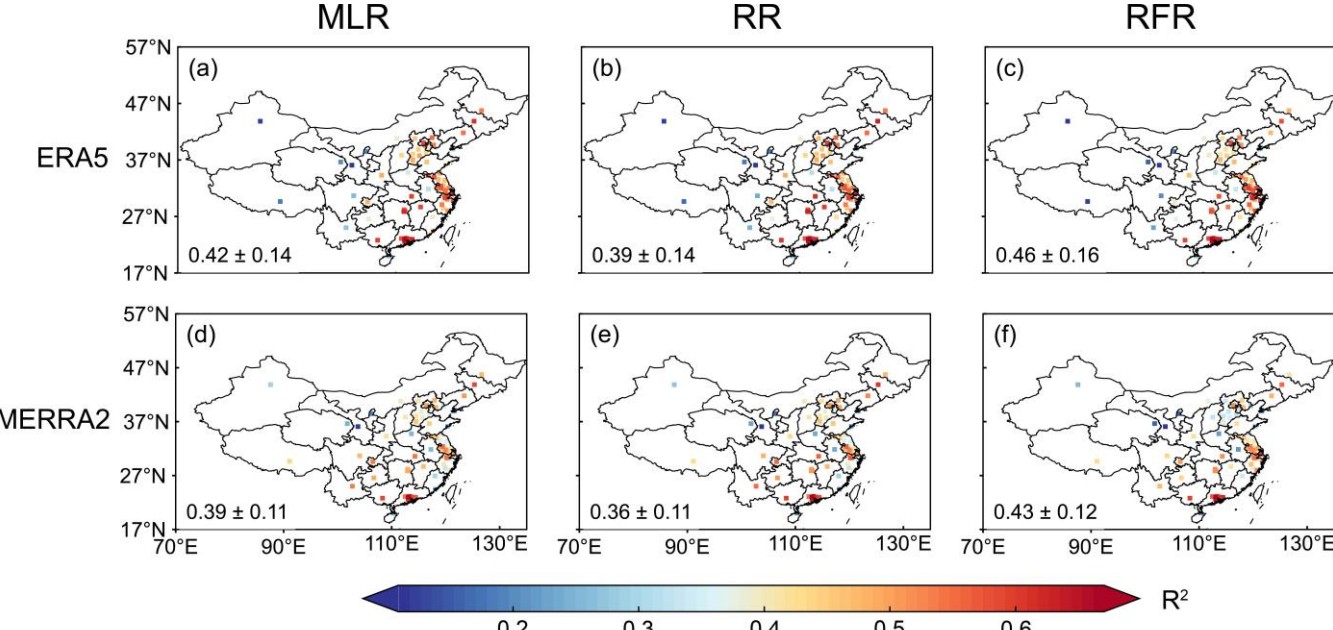

Figure 13: Predictive skills of daily MDA8 ozone at Chinse cities during June to August from 2013 to 2019 by the multiple linear regression (MLR) method, the ridge regression (RR) method, and the random forest regression (RFR) methods, using the meteorological parameters from two re-analysis dataset (ERA5 and MERRA-2). The predictive skills are presented by the coefficient of determination ($R^2$). Values are the mean ± standard deviation across 74 cities. See Section 2.2 for the descriptions of the three models.



**Figure 14: Trends in observed summertime ozone and model-derived meteorological driven trends key city-clusters in China. Results are MDA8 ozone anomalies for individual JJA months averaged over the cities relative to the 2013–2019 mean. Observed trends are compared to the meteorologically driven trends diagnosed by the MLR, RR, and RFR models, using both ERA5 and MERRA2 meteorological data. The observed trends and mean estimates of the trends driven by meteorology and emissions averaged over the three methods are shown inset, with the trends estimated by individual methods shown in the parentheses.**





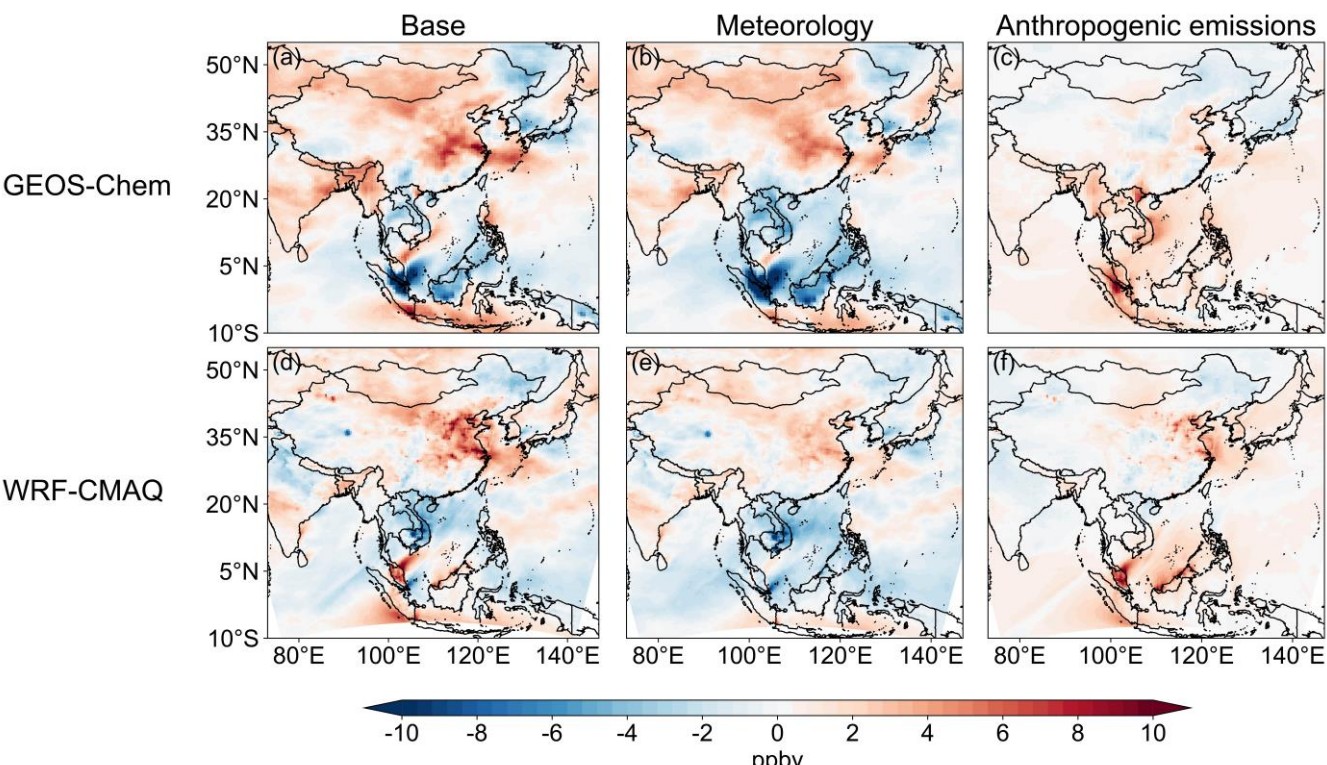

**Figure 15: Differences in simulated June–August mean MDA8 ozone concentrations between 2017-2019 and 2013-2015**
1510 **(2017-2019 minus 2013-2015). Panel (a), (b), and (c) shows the difference simulated surface MDA8 ozone concentration, contribution from meteorological conditions, and contribution from anthropogenic emissions, respectively, from the GEOS-Chem model. Panels (d), (e), and (f) are the same as (a), (b), and (c) but for the WRF-CMAQ model.**





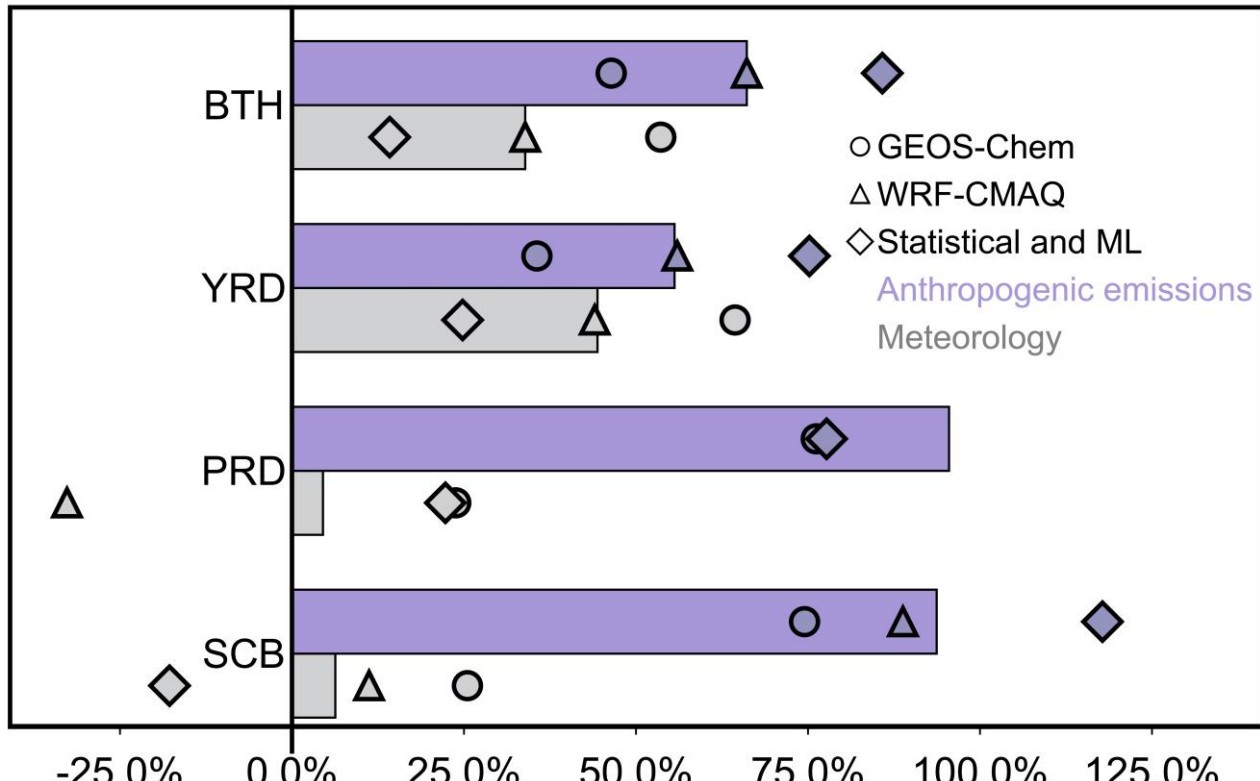

Figure 16: Summary of the meteorology-driven and anthropogenic emission-driven surface MDA8 ozone difference between 2017-2019 and 2013-2015 from the statistical model, machine learning model, and chemical models. Results are summarized from Table 3, adding the estimates from statical and machine learning model using ERA5 as input. See the text for the detailed calculation.