# Peer review of "Tropospheric ozone trends and attributions over East and Southeast Asia in 1995-2019: An integrated assessment using statistical methods, machine learning models, and multiple chemical transport models"

_EGUsphere, 2024_

## Author Comment (AC1)

**Reviewer 1: Dr. I. Pérez**

**[Comment #1-1]:** This is a quite complete paper about tropospheric ozone in East and Southeast Asia in the past 25 years. Many authors are involved, and the study considers an extensive station network and modelling studies where 80% of data are for training and 20% of data are for testing. The analysis is focused on the summertime and surface concentrations, profiles and trends are investigated. Finally, the contribution of anthropogenic emissions and meteorology are quantified. Due to the extension and intensity of this analysis, it merits to be published in Atmospheric Chemistry and Physics, although the following minor issues should be answered by the authors.

**[Response #1-1]: We thank Dr. I. Pérez for the positive and valuable comments. All of them have been implemented in the revised manuscript. Please see our itemized responses below.**

**[Comment #1-2]:** Some results could be mixed with the discussion in the current paper, due to their comparison with other studies. Perhaps the authors could indicate if both sections could be more separated.

**[Response #1-2]: Thank you for your comment. In Sections 3, 4, and 5, we mainly focus on the interpretation of the results, incorporating comparisons with other relevant studies or supporting evidence from references where appropriate. We believe this structure helps readers contextualize the findings of this study within the broader literature. Section 5.3 is dedicated to an independent discussion of the mechanisms driving the summertime surface ozone increase in China from 2013 to 2019, drawing extensively on existing references. Section 6 is organized into two parts: the first half summarizes the key findings of this study, the second half addresses limitations and future directions. We hope this structure provides a balanced and coherent presentation of both the results and discussions.**

**[Comment #1-3]:** Limitations of this study could be highlighted. Moreover, a comment about results extrapolation to the future would be acknowledged by the readers.

**[Response #1-3]: Thank you for pointing it out. We have added the following discussion in Section 6. "_The quantitative ozone response to precursor emissions and climate change, as simulated by multi-models in this study, holds significant implications for future ozone projections. In the free troposphere, our results have shown that ozone changes largely aligned with trends in $NO_x$ emissions over the ESEA. In the future, continued reductions in $NO_x$ and VOCs emissions in China are expected to further decrease its contribution to global tropospheric ozone burden (Han et al., 2024). At the surface, although emission control measures since 2013 have contributed to ozone enhancement in China, they are projected to reduce ozone as emission reductions deepen (Li et al., 2019c; Lu et al., 2021a). In other parts of ESEA, while future emission scenarios will be highly dependent on policy decisions, it is apparent that emissions from Southeast Asia will significantly affect both local ozone air quality and global tropospheric ozone burden due to the high efficiency of ozone chemical production and strong vertical transport in this region. Our simulations also capture the positive response of surface ozone concentrations over the ESEA to global warming. In consistent, multiple model results predict that the positive slope of surface ozone concentration with increasing temperature (also known as the ozone climate penalty) will persist in this heavily polluted region under future scenarios, in contrast to the ozone decrease in remote regions (Zanis et al., 2022), although such penalty effect is expected to diminish as emission reductions progress (Chang et al., 2024; Li et al., 2025). The ozone_**

*climate penalty effect requires further reduction in anthropogenic emissions of ozone precursors.***"**

**Reference**

Chang, K.-L., McDonald, B. C., and Cooper, O. R.: Surface ozone trend variability across the United States and the impact of heatwaves (1990–2023), Atmospheric Chemistry and Physics Discussions, https://doi.org/10.5194/egusphere-2024-3674, 2024.

Han, H., Zhang, L., Wang, X., and Lu, X.: Contrasting Domestic and Global Impacts of Emission Reductions in China on Tropospheric Ozone, J. Geophys. Res., 129, https://doi.org/10.1029/2024jd041453, 2024.

Li, S., Wang, H., and Lu, X.: Anthropogenic emission controls reduce summertime ozone–temperature sensitivity in the United States, Atmos. Chem. Phys., 25, 2725–2743, https://doi.org/10.5194/acp-25-2725-2025, 2025.

Li, K., Jacob, D. J., Liao, H., Zhu, J., Shah, V., Shen, L., Bates, K. H., Zhang, Q., and Zhai, S.: A two-pollutant strategy for improving ozone and particulate air quality in China, Nature Geosci., https://doi.org/10.1038/s41561-019-0464-x, 2019c.

Lu, X., Ye, X., Zhou, M., Zhao, Y., Weng, H., Kong, H., Li, K., Gao, M., Zheng, B., Lin, J., Zhou, F., Zhang, Q., Wu, D., Zhang, L., and Zhang, Y.: The underappreciated role of agricultural soil nitrogen oxide emissions in ozone pollution regulation in North China, Nature Communications, 12, https://doi.org/10.1038/s41467-021-25147-9, 2021a.

Zanis, P., Akritidis, D., Turnock, S., Naik, V., Szopa, S., Georgoulias, A. K., Bauer, S. E., Deushi, M., Horowitz, L. W., Keeble, J., Le Sager, P., O'Connor, F. M., Oshima, N., Tsigaridis, K., and van Noije, T.: Climate change penalty and benefit on surface ozone: a global perspective based on CMIP6 earth system models, Environmental Research Letters, 17, https://doi.org/10.1088/1748-9326/ac4a34, 2022.

**[Comment #1-4]:** Minor remarks.

L. 367. "aaply" or "apply"?

L. 621. "Aisa" or "Asia"?

**[Response #1-4]: Corrected as suggested.**

---

## Author Comment (AC2)

**Reviewer 2:**

**[Comment #2-1]:** This study investigates tropospheric ozone trends and attributions over East and Southeast Asia from 1995 to 2019 using an integrated approach that includes statistical methods, machine learning models, and multiple chemical transport models (CTMs). The paper provides valuable insights into the key drivers of ozone changes, including anthropogenic emissions and meteorological influences. The methodology is sound.

However, the manuscript has several major issues, particularly regarding the uncertainty quantification of models, statistical method validation, the role of meteorological factors, and the clarity of explanations. Below, I provide detailed comments with specific suggestions for revision. Once these revisions are addressed, the manuscript will provide a stronger contribution to understanding ozone trends in East and Southeast Asia.

**[Response #2-1]: We thank the reviewer for the positive and valuable comments. All of them have been implemented in the revised manuscript. Please see our itemized responses below.**

**[Comment #2-2]:** Page 4, Line 60: "Ozone is growing especially fast over the densely populated regions of East and Southeast Asia." "Ozone concentrations are increasing rapidly over..."
**[Response #2-2]: Corrected as suggested.**

**[Comment #2-3]:** Page 7, Lines 170-180: "We adopt one conventional statistical method, i.e., the multiple linear regression (MLR) method, and two machine learning models, i.e., the ridge regression (RR) and random forest regression (RFR) methods." Comment: These methods are useful, but have their assumptions been checked? For example: MLR: Have you tested for multicollinearity e.g. Variance Inflation Factor, VIF? RR: Why was Ridge Regression chosen over Lasso Regression, which can also reduce collinearity? RFR: Decision tree-based models can be prone to overfitting, especially when trained on high-dimensional meteorological variables.

**[Response #2-3]:**
**We thank the reviewer for their insightful questions.**
**(1) MLR: Have you tested for multicollinearity e.g. Variance Inflation Factor, VIF?**
**We have calculated the variance inflation factor (VIF) values for MLR model as suggested. Among the 74 cites, we find that 68 cities (over 90%) show low VIF values (<5 for all predictors), with only 6 cities—mostly located in west of eastern China (out of the four city clusters)—showing relatively higher VIF (above 5) for wind variables such as U10 (zonal wind at 10m), U (zonal wind at 850hPa) from ERA5. This indicates that multicollinearity is not a prevalent issue across cities. Furthermore, we also conducted a test run using MLR with all 11 meteorological predictors included. Its predictive skills—as estimated by $R^2$—across these cities remain similar to the less-prone-to-multicollinearity RR, suggesting that the effect of multicollinearity is limited.**

**Nonetheless, we have made updates to the MLR reported in the original manuscript to ensure it uses a backward stepwise method that consistently retains 5 predictors in the final model for all cities. This change is based on further testing, which showed that this setup provides the most optimal balance between achieving satisfactory predictive performance and avoiding overfitting. Additionally, the updated MLR now strictly follows the same training-testing split as the two machine learning algorithms—RR and RFRs—allowing a more**

objective comparison across these three algorithms.

We have revised the text in Section 2.2. *"We use a backward stepwise MLR modeling approach, starting with all 11 meteorological variables (see below) as predictors and iteratively remove the least significant ones until five remain. MLR then models only relying on these five predictors, thereby reducing potential collinearity and the risk of overfitting that are often associated with conventional MLR, in which all predictors are considered. We also apply the variance inflation factors (VIF; the inverse of tolerance) to measure the collinearity of the MLR models."*

In addition, figures and tables reporting results (Figures 13, 14, 16; Table4, etc.) from the MLR models have been updated accordingly.

**(2) RR: Why was Ridge Regression chosen over Lasso Regression, which can also reduce collinearity?**

The difference between Ridge Regression and Lasso Regression lies in their regularization methods, with L2 regularization used in Ridge Regression and L1 in Lasso Regression. With the augmentation of L1-regularization, Lasso Regression might encourage solution of zero slopes for certain predictors (i.e., omitting predictors) that are considered to be less important. In this study, we intend to include an algorithm that retains all predictors while ensuring the model can handle potential collinearity and reduce the risk of overfitting. Ridge regression with L2-regularization is therefore preferable.

**(3) RFR: Decision tree-based models can be prone to overfitting, especially when trained on high-dimensional meteorological variables.**

A single decision tree is likely prone to overfitting, as it often struggles to achieve an optimal bias-variance tradeoff, typically due to suboptimal hyperparameter settings. While Random Forest Regression (RFR), combining "bagging" and "random subset of features", is less susceptible to this issue.

"Bagging" involves "bootstrap sampling" and "aggregation". Specifically, each tree within the forest is trained on a random subset of data, which allows the regression trees to be more diverse. The limitation is that these subsets may overlap (i.e., sharing same data points) because of the nature of "bootstrap sampling". To counter this, RFR adopts "random subset of features", in which each regression tree is learned on a subset of features, further diversifying regression tress and thereby reducing the impact of high dimensionality and further minimizing overfitting.

While a number of individual trees may still under- or overfit, the final aggregation across the forest helps mitigate these issues. Detailed description of the ran can be referred to (Altman & Krzywinski, 2017).

We have added the following discussion in Section 2.1 to reflect this point "*RFR, on the other hand, is an ensemble decision tree approach that can adaptively model both linear and nonlinear relationships between predictors and the dependent variable (Breiman, 2001; Grange et al., 2018). Its use of bootstrap sampling and random feature subsets in each regression tree makes it more resistant to overfitting than a single tree prediction (Altman and Krzywinski, 2017).*"

**Reference**

Altman, N. and Krzywinski, M.: Ensemble methods: bagging and random forests, Nat. Methods, 14, 933-934, https://doi.org/10.1038/nmeth.4438, 2017.

**[Comment #2-4]:** Page 8, Lines 200-210: "We follow standard machine learning practices by splitting the training and testing sets for RR and RFR." Comment: It is unclear whether time-series dependencies were considered when splitting the dataset. Standard random splitting may not be appropriate for time-dependent ozone trends.

**[Response #2-4]:**

We thank the reviewer for raising the possibility of temporal autocorrelation i.e., time-series dependencies due to the standard random splitting method. To assess its potential impact on model performance and the quantification of meteorologically driven ozone, we conduct a test using ridge regression, excluding three days of training data adjacent to the test data. For instance, when predicting MDA8 ozone for 15-July-2015, training data from 12–14 July and 16–18 July are discarded. This creates a buffer between training and test data, reducing the impact of temporal autocorrelation. We then compare this run (termed as RR_r3d) with the default ridge regression (RR) model to evaluate the effect of autocorrelation. The results show that the predictive skills and meteorological quantifications stay similar, suggesting limited impact from temporal autocorrelation in this study.

[Figure]

Figure R1. Comparison of predictive skills, measured by the coefficient of determination ($R^2$), between the default ridge regression (RR) as described in the main text of the manuscript and the ridge regression, with three days of training data adjacent to the test data (termed as RR_r3d) excluded. The average value of $R^2$ with its standard deviation across 74 cities are shown in the bottom left corner of each panel.

[Figure]

Figure R2. Comparison of meteorologically driven ozone trends derived by RR and RR_r3d (see caption from Figure. R1) in the four key city clusters in China.

**[Comment #2-5]:** Page 8, Line 200: "We follow standard machine learning practices by splitting the training and testing sets for RR and RFR." – misleading
"We follow standard machine learning practices by splitting the dataset into training and testing sets for RR and RFR."

**[Response #2-5]: Corrected as suggested. As mentioned in the response above, we reflect the application of the same training-testing split on MLR in the main text:**

_**"We follow standard machine learning practices by splitting the dataset into training and testing sets for MLR, RR and RFR."**_

**[Comment #2-6]:** Page 13, Lines 395-400: "A notable upward trend in surface downward solar radiation is discernible across Southeast Asia, whereas a decline is evident in most parts of China." Comment: This statement is correct but lacks explanation. The decrease in solar radiation over China is likely linked to aerosol reductions, while increases in Southeast Asia may be due to decreasing cloud cover. Were aerosol-cloud interactions important here?

**[Response #2-6]: Thank you for pointing it out. We have analyzed the cloud cover fraction and find that changes in surface downward solar radiation are associated align with trends in total**

cloud coverage. The decrease in eastern China is partly attributed to the increase in aerosol as pointed by previous studies (e.g. He et al., 2018). We have added the following discussions in the text *"A notable upward trend in surface downward solar radiation is discernible across Southeast Asia, whereas a decline is evident in most parts of China. These shifts in solar radiation align with trends in total cloud coverage. The decline in surface downward solar radiation in eastern China is also attributable to increase in aerosol loading (He et al., 2018)."*

**Reference**

He, Y., Wang, K., Zhou, C., and Wild, M.: A Revisit of Global Dimming and Brightening Based on the Sunshine Duration, Geophys. Res. Lett., 45, 4281-4289, https://doi.org/10.1029/2018gl077424, 2018.

**[Comment #2-7]:** Page 13, Lines 405-415: "Significant positive trends in BVOC emissions are shown in eastern China, Southeast Asia, and parts of India, in contrast to a significant decline in Myanmar." Comment: Why Myanmar is different?

**[Response #2-7]: We have revised the text for clarification** *"Figure S5 also illustrates the temporal trends in these natural emissions. Significant positive trends in BVOC emissions are shown in eastern China, Southeast Asia, and parts of India, in contrast to a significant decline in Myanmar. This is most likely driven by temperatures, which rise in most regions in ESEA but decrease in Myanmar (Figure S4b)."*

**[Comment #2-8]:** Page 14, Lines 450-460: "We find that, overall, all models applied in this study capture the observed ozone vertical profiles over East Asia and Southeast Asia." Comment: From my perspective, the large-scale vertical distribution of simulated ozone should not vary significantly across different models. Therefore, simply providing an average bias difference of a few ppb is clearly insufficient to indicate good model performance. Moreover, the discrepancies at the surface are excessively large in Southeast Asia.

**[Response #2-8]: We agree. We have added a scatter plot with the correlation coefficient inset in the Supplementary Information. We have also added the following discussion in the text** *"Overall, we find that the GC05 model shows better agreement with the observed distribution in tropospheric ozone compared to the WRF-CMAQ model, as indicated by the higher correlation coefficients and smaller relative bias to observations (Fig. S6)."*.

[Figure]

**Figure S6**. Evaluation of GEOS-Chem and WRF-CMAQ simulated summertime tropospheric ozone distributions over IAGOS regions and at ozonesonde sites. The comparisons are separated for the upper, middle, and lower troposphere. Correlation coefficients between the observed and simulated values are shown inset. Data descriptions are provided in the caption of Figure 3.

**[Comment #2-9]:** Page 15, Lines 470-475: "Overall, all models capture the spatial distributions of surface ozone over ESEA, as indicated by the high spatial correlation coefficients between the observed and simulated values ranging from 0.50-0.78." Comment: While spatial correlation coefficients provide some measure of agreement, they do not reveal the absolute differences in ozone levels. I suggest including scatter plots to compare observed vs. modeled values more rigorously.

**[Response #2-9]: We agree. We have followed your suggestion to add a scatter plot to compare observed vs. modeled values in Figure S7. We have stated in the text _"Overall, all models capture the spatial distributions of surface ozone over ESEA, as indicated by the high spatial correlation coefficients between the observed and simulated values ranging from 0.50-0.78 (except for Thailand where only 11 sites are available), but they tend to overestimate surface ozone concentrations over ESEA, as also indicated in Figure S7."_**

[Figure]

**Figure S7**. Evaluation of GEOS-Chem and WRF-CMAQ model simulated summertime surface MDA8 ozone concentrations over ESEA. Correlation coefficients between the observed and simulated values are shown inset. Same as Figure 4 but illustrated in scatter plots. CAM-Chem results are not shown as the spatial resolution is too coarse to resolve the ozone deviation at different sites.

**[Comment #2-10]:** The y-axis values in Figure 6 are too large; a range from 0 to 100 is not appropriate. In most regions, ozone changes are relatively flat.

**[Response #2-10]: We agree. We have adjusted the range to 0-80 ppbv. We hope to maintain consistency in the range across different regions, thereby preventing potential misinterpretations when comparing ozone concentrations between various regions.**

[Figure]

**Figure 6.** Evaluation of GEOS-Chem, WRF-CMAQ, and CAM4-chem model ability to capture 1995-2019 summertime surface ozone trends. The selected periods for each region/site correspond to the overlapping years of available observations and simulations. Ozone trends derived from the observations and different models, the associated p-values, and the correlation coefficients between the observed and simulated values are shown inset.

**[Comment #2-11]:** In Figure 7, the term "other anthropogenic" is unclear; it actually refers to emissions from outside ESEA. Additionally, regarding the impact of climate change, the two models exhibit different spatial differences, including for surface ozone in Figure 11. The regional discrepancies should be further explained by identifying the specific climate changes responsible for these variations.

**[Response #2-11]: Thank you for pointing it out. We have revised the terms to "Global anthropogenic emissions (including CH₄+Aircraft)", "Anthropogenic emissions from ESEA"**

and **"Anthropogenic emissions outside ESEA"**. **Pleased find the revised Figure 7, Figure 8, and Figure 11.**

**Regarding the specific climate factors responsible for the difference between the two models, we have added following discussions in Section 5.1.2** *"The influence of climate change on surface ozone also shows significant interannual variability, as well as variations across the GEOS-Chem and WRF-CMAQ model that could be attributed to the differences in meteorological variables used to drive the chemic model. For example, The GEOS-Chem model simulates a larger surface ozone enhancement from 1995 to 2019 over the eastern China than WRF-CMAQ (Figure 11(e3) vs Fig. 11(e6)), which can be explained by the larger increase in temperature and decrease in wind speed from the MERRA2 reanalysis dataset than the WRF simulation (Fig.S11). Difference in temperature, solar radiation, and specific humidity between the two meteorological fields can also explain the climate-driven ozone difference between GEOS-Chem and WRF-CMAQ over Malaysia and surrounding areas."*

[Figure]

**Figure S11.** Differences in meteorological factors between the corresponding year and 1995 from the MERRA2 reanalysis data used to drive the GEOS-Chem model and the WRF model output used to drive the WRF-CMAQ model.

**[Comment #2-12]:** Another point regarding the tropospheric ozone column changes in Figure 7: GEOS-Chem does not exhibit strong sensitivity to individual variables. Why does its change in the Base experiment appear comparable to that of WRF-CMAQ? This needs further clarification.

**[Response #2-12]: Thank you for pointing it out. Indeed, we find that the variability of tropospheric ozone column (TCO) driven by changes in anthropogenic emissions of GEOS-Chem is smaller than that of WRF-CMAQ. However, the TCO variability driven by climate change in GEOS-Chem is larger. Therefore, the TCO change in the Base experiment is comparable between the two models. The difference in the attribution may reflect the distinct characteristics in the two models. We have added the following discussion on this issue in the end of Section 5.1.1.**

*"While both the GEOS-Chem and WRF-CMAQ model demonstrate consistent increase of tropospheric ozone burden and their attribution from 1995 to 2019 (53% in GEOS-Chem and 59% in WRF-CMAQ attributable to anthropogenic emissions, 47% in GEOS-Chem and 41% in WRF-CMAQ attributable to climate change), differences in the magnitude and regional responses reflect the distinct characteristics between the two models. GEOS-Chem tends to attribute a larger portion of ozone change to climate factors, including its influence on STE and natural emissions. This is partly due to GEOS-Chem's detailed representation of stratospheric chemistry, whereas WRF-CMAQ does not explicitly simulate stratospheric chemistry but instead applies chemical boundary conditions as inputs. Additionally, GEOS-Chem incorporates a parameterization for lightning $NO_x$ emissions based on cloud-top height, a feature absent in WRF-CMAQ. Furthermore, the two models utilize different meteorological fields to drive their chemical modules. These factors likely contribute to their differing attributions of tropospheric column ozone (TCO) changes to climate. Regarding the attribution to emissions, although both models show similar trends in total anthropogenic emissions of ozone precursors over ESEA (Table 3), differences in spatial resolution and chemical mechanisms are expected to influence their respective contributions to ozone. These discrepancies underscore the importance of employing multiple chemical models to quantify ozone trend attributions robustly."*

**[Comment #2-13]:** The description of OPE (Ozone Production Efficiency) is too brief. Would it be worthwhile to include a separate figure for it? The authors have already used $H_2O_2/HNO_3$ as an indicator, which should be sufficient. However, in Figure 12, the color bar is unclear, making it difficult to interpret how this indicator changes.

**[Response #2-13]: We agree that the discussion of OPE (Ozone Production Efficiency) is too brief and that the discussion of $H_2O_2/HNO_3$ should have been sufficient to demonstrate the OPE. In addition, WRF-CMAQ model does not output OPE. For these reasons, we have removed the figure of OPE in Figure 12 and associated discussions from the text.**

**[Comment #2-14]:** Figure 13 presents the predictive performance of different statistical learning models. However, R² alone is not sufficient for evaluation. The key aspect here is the model error, i.e., how much of the ozone variation can actually be explained by the model, rather than just the correlation.

**[Response #2-14]: We agree, and $R^2$ is exactly the statistical metrics used to measure the**

proportion of the variance in daily ozone concentrations that is explained by the meteorological parameters in the model. We note that our purpose is not to build a perfect statistical model for ozone based on meteorological variables that strives for the highest R² or low lowest biases. Instead, we aim to isolate the portion of ozone variability that can be explained by meteorological variables. As such, we do not provide the mean-biases (MB) or normalized mean biases (NMB) s to avoid misleading. We have added the following text in Section 5.2.1 for clarification. *"Figure 13 summarizes the coefficient of determination ($R^2$) from all these algorithms using two sets of meteorological data. $R^2$ measures the proportion of the variance in daily ozone concentrations that is explained by the meteorological parameters in the model. It should be noted that our purpose is not to build a model that perfectly reconstructs observational ozone variabilities (i.e., striving for the highest R² and lowest biases), as this is not feasible given that the meteorological variables are the sole predictors in this model framework. Instead, we aim to estimate the portion of ozone variability that can be explained by meteorological variables. "*

**[Comment #2-15]:** From Figure 14, it is evident that the influence of meteorology in all models appears relatively flat compared to the observed values. This suggests that the observed ozone variations cannot be fully explained by meteorology alone. Here, the emission impact is calculated as the observed values minus the meteorology-driven component, i.e., the residual. How exactly are the emission-driven values calculated? This should be clearly explained in the figure description. However, if we instead subtract the emission-driven values from the observed values, would the resulting values be exactly equal to the meteorology-driven component shown in Figure 14?

**[Response #2-15]: Thank you for pointing it out. We have clarified in the text and also in the figure caption of Figure 14:** *"Here, we quantify ozone trends attributable to meteorological factors by calculating the linear trends of ozone concentrations predicted exclusively from meteorological variables using statistical or machine learning models. We then obtain the ozone residual by subtracting the meteorologically predicted ozone from the observed ozone concentrations. The trends of these residuals are interpreted as ozone trends driven by changes in anthropogenic emissions. This strategy follows previous study Li et al. (2019b) and Weng et al. (2022)."* **In this way, subtracting the emission-driven values from the observed values would be exactly equal to the meteorology-driven trends.**

**[Comment #2-16]:** It would be helpful to include an ensemble mean of all statistical and physical models in Figure 16. Since all models inherently have biases and uncertainties, an average would help mitigate individual model errors.

**[Comment #2-16]: We agree. In the manuscript, we have shown the ensemble mean of all statistical and physical models with the horizontal bars in Figure 16, but they are not clearly indicated in the figure caption. We have now added the following text in the figure caption for clarification** *"The circles and triangles represent result from GEOS-Chem and WRF-CMAQ, respectively. The diamond symbols represent the average of six estimates from the statistical and machining learning model, including three models (MLR, RR, and RFR) with predictions from two meteorological inputs (MERRA2 and ERA5) in turn. Horizontal bars then represent the mean of the statistical and machining learning model (the ensemble mean from six estimate), GEOS-Chem, and WRF-CMAQ model."*